# Polyunsaturated fatty acid analogues differentially affect cardiac Na$_V$, Ca$_V$, and K$_V$ channels through unique mechanisms

Briana M Bohannon[1†], Alicia de la Cruz[1†], Xiaoan Wu[1], Jessica J Jowais[1], Marta E Perez[1], Derek M Dykxhoorn[2], Sara I Liin[3], H Peter Larsson[1*]

[1]Department of Physiology and Biophysics, Miller School of Medicine, University of Miami, Miami, United States; [2]John P. Hussman Institute for Human Genomics, University of Miami Miller School of Medicine, Miami, United States; [3]Department of Biomedical and Clinical Sciences, Linköping University, Linköping, Sweden

**Abstract** The cardiac ventricular action potential depends on several voltage-gated ion channels, including Na$_V$, Ca$_V$, and K$_V$ channels. Mutations in these channels can cause Long QT Syndrome (LQTS) which increases the risk for ventricular fibrillation and sudden cardiac death. Polyunsaturated fatty acids (PUFAs) have emerged as potential therapeutics for LQTS because they are modulators of voltage-gated ion channels. Here we demonstrate that PUFA analogues vary in their selectivity for human voltage-gated ion channels involved in the ventricular action potential. The effects of specific PUFA analogues range from selective for a specific ion channel to broadly modulating cardiac ion channels from all three families (Na$_V$, Ca$_V$, and K$_V$). In addition, a PUFA analogue selective for the cardiac I$_{Ks}$ channel (Kv7.1/KCNE1) is effective in shortening the cardiac action potential in human-induced pluripotent stem cell-derived cardiomyocytes. Our data suggest that PUFA analogues could potentially be developed as therapeutics for LQTS and cardiac arrhythmia.

*For correspondence:
plarsson@med.miami.edu

†These authors contributed equally to this work

## Introduction

The human ventricular action potential is mediated by the coordinated activity of several different voltage-dependent ion channels (*Mohrman and Heller, 2010*). The ventricular myocyte is hyperpolarized during diastole between two action potentials mainly due to inward rectifier I$_{K1}$ K$^+$ channels. The ventricular action potential spreads between cardiomyocytes by currents through gap junction channels. The rapid upstroke of the ventricular action potential is mediated by the activation of the voltage-gated Na$^+$ channel, Nav1.5, which then rapidly inactivates. This is followed by the activation and inactivation of the voltage-gated I$_{to}$ K$^+$ channels. The activation of L-type voltage gated Ca$^{2+}$ channels, Cav1.2, and influx of Ca$^{2+}$ leads to a sustained depolarization, or plateau phase, and the contraction of the cardiac muscle. Inactivation of Cav1.2 channels along with the activation of the slow delayed-rectifier K$^+$ channels, Kv11.1 (which generates the I$_{Kr}$ current) and Kv7.1/KCNE1 (which generates the I$_{Ks}$ current), work to promote repolarization of the cell membrane together with the I$_{K1}$ current (*Nerbonne and Kass, 2005*). Mutations of these ion channels (or channelopathies) could lead to Long QT Syndrome (LQTS), which is an arrhythmogenic disorder that predisposes the individual to potentially fatal cardiac arrhythmias (*Alders and Christiaans, 2003*; *Bohnen et al., 2017*). We will here focus on the effects of PUFA analogues on Nav1.5, Cav1.2, and Kv7.1 together with some of their associated subunits to try to assess the effects of PUFA analogues on the currents through these channels on the ventricular action potential.

The Nav1.5 α-subunit contains four non-identical linked domains, domains I-4 (DI-DIV). Each of these domains contain 6 transmembrane segments (S1-S6), where the S1-S4 segments make up the

voltage-sensing domains (VSD) and S5-S6 segments make up the pore domains (PD). The S4 helix of each of the four domains contains a motif with positively charged amino acid residues which allow the S4 segment to detect and respond to changes in the membrane electric field, acting as the channel voltage sensor (*Chanda and Bezanilla, 2002*). The movement of these voltage sensors determines the voltage dependence of activation and inactivation, where activation of the DI-III S4s are suggested to promote channel activation and activation of DIV S4 segment is sufficient to induce voltage-dependent inactivation of Nav1.5 (*Capes et al., 2013*). The Nav1.5 α-subunit exists as a macromolecular complex with the accessory subunit β1. β1 is a single transmembrane spanning helix that modifies the kinetics of Nav1.5 channel activation and inactivation and can alter the pharmacology of the Nav1.5 α-subunit (*Barro-Soria et al., 2017*; *Xiao et al., 2000*; *Zhu et al., 2017*). Gain-of-function mutations of Nav1.5 increase $Na^+$ currents and lead to LQTS Type 3 (LQT3) (*Fernández-Falgueras et al., 2017*; *Calloe et al., 2013*; *Rivolta et al., 2002*).

Like Nav1.5, the Cav1.2 α-subunit contains four linked domains, DI-DIV, where each domain consists of 6 transmembrane segments S1-S6. S1-S4 form the VSD, where S4 acts as the voltage sensor, and S5-S6 form the PD. Cav1.2 exists as a large macromolecular complex with the accessory subunits β3 and α2δ subunits that are important for membrane expression and alter channel activation and deactivation kinetics, respectively (*Rougier and Abriel, 1863*; *Chen et al., 2004*). Cav1.2/β3/α2δ undergoes both voltage-dependent inactivation and calcium-dependent inactivation (*Stotz et al., 2004*; *Zhang et al., 1994*) which allows it to regulate $Ca^{2+}$ influx into the cardiomyocyte. Gain-of-function mutations of Cav1.2 increase $Ca^{2+}$ currents and lead to LQT Type 8 (LQT8) (*Dick et al., 2016*; *Hoffman, 1995*).

The voltage-gated $K^+$ channel, Kv7.1, along with the auxiliary subunit KCNE1, mediates an important repolarizing $K^+$ current, $I_{Ks}$ (*Noble and Tsien, 1969*; *Deal et al., 1996*; *Lei and Brown, 1996*). The Kv7.1 α-subunit contains 6 transmembrane segments, S1-S6. S1-S4 comprise the VSD, where S4 contains several positively charged amino acid residues that allow S4 to act as the voltage sensor of Kv7.1. S5-S6 segments comprise the channel PD. Kv7.1 forms a tetrameric channel, where 4 Kv7.1 α-subunits arrange to form a functional channel. The auxiliary β-subunit KCNE1 drastically modulates Kv7.1 channel voltage dependence, activation kinetics, and single-channel conductance (*Barro-Soria et al., 2014*; *Osteen et al., 2010*). Loss-of-function mutations in the Kv7.1 α-subunit and KCNE1 β-subunit lead to reductions in $I_{Ks}$ and can lead to LQTS Type 1 (LQT1) and Type 5 (LQT5) (*Huang et al., 2018*; *Ma et al., 2015*; *Schwartz et al., 2012*; *Sanguinetti, 1999*; *Harmer et al., 2010*), respectively.

Polyunsaturated fatty acids (PUFAs) are amphipathic molecules that have been suggested to possess antiarrhythmic effects (*Endo and Arita, 2016*; *Kang and Leaf, 2000*). PUFAs are characterized by having a long hydrocarbon tail with two or more double bonds, as well as having a charged, hydrophilic head group (*Benatti et al., 2004*). PUFAs, such as DHA and EPA, have been shown to prevent cardiac arrhythmias in animal models and cultured cardiomyocytes by inhibiting the activity of $Na_V$ and $Ca_V$ channels (*Kang and Leaf, 2000*; *Kang and Leaf, 1996*; *Xiao et al., 1997*; *Xiao et al., 1995*). Specifically, DHA and EPA are thought to bind to discrete sites on the channel protein to stabilize the inactivated states of the $Na_V$ and $Ca_V$ channels (*Kang and Leaf, 1996*; *Xiao et al., 2001*). Since the voltage sensors of $Na_V$ and $Ca_V$ channels are relatively homologous, it has been suggested that PUFAs act on the voltage-sensing S4 segments that control inactivation in these channels (*Kang and Leaf, 2000*; *Kang and Leaf, 1996*). Our group has demonstrated that PUFAs and PUFA analogues also modulate the activity of the Kv7.1/KCNE1 channel and work to promote voltage-dependent activation of the $I_{Ks}$ current (*Börjesson et al., 2008*; *Liin et al., 2015*; *Liin et al., 2016*). The mechanism through which PUFAs promote Kv7.1/KCNE1 activation is referred to as the lipoelectric hypothesis which involves the following: 1) the PUFA molecule integrates into the membrane via its hydrocarbon tail and 2) the negatively charged PUFA head group electrostatically attracts the positively charged S4 segment and facilitates the outward movement of S4, promoting Kv7.1/KCNE1 channel activation (*Börjesson et al., 2008*; *Bohannon et al., 2020*). Our group has also demonstrated that PUFAs and modified PUFAs exert a second effect on the pore of Kv7.1 through an additional electrostatic interaction with a lysine residue (K326) in the S6 segment (*Liin et al., 2018a*). This electrostatic interaction between the negatively charged PUFA head group and K326 leads to an increase in maximal conductance of the channel (*Liin et al., 2018a*). In addition, previous investigation of the anti-arrhythmic potential of PUFA analogues have shown that the

negatively charged PUFA analogues are able to shorten a pharmacologically prolonged ventricular action potential and QT interval in isolated guinea pig hearts (*Liin et al., 2015*).

In previous experiments, it was discovered that activation of Kv7.1 by DHA is abolished when Kv7.1 is co-expressed with the auxiliary subunit KCNE1, which is necessary for the physiological $I_{Ks}$ current (*Liin et al., 2015*). However, PUFA-induced activation of the cardiac Kv7.1/KCNE1 channel can be restored by conducting experiments using DHA at pH 9 (*Liin et al., 2015*). Recent work has demonstrated that KCNE1 induces a conformational change in Kv7.1 that bring acidic residues (namely E290) closer to the PUFA binding site, leading to protonation of the PUFA head group (*Larsson et al., 2018*). The deprotonated, negatively charged head group is necessary for PUFA-induced activation of the Kv7.1/KCNE1 channel. Therefore it is necessary to use modified PUFAs that are deprotonated at physiological pH (7.5) (*Liin et al., 2015*). We found that the PUFA analogues N-arachidonoyl taurine (N-AT) and docosahexaenoyl glycine (DHA-glycine) have head groups with a lower pKa, allowing these analogues to remain negatively charged at physiological pH and activate the Kv7.1/KCNE1 channel (*Liin et al., 2015*; *Liin et al., 2016*; *Bohannon et al., 2020*).

Some groups have suggested that PUFAs could modify $Na_V$ channels by causing a leftward shift in voltage dependent inactivation through an electrostatic effect on the voltage-sensing domains involved in inactivation (*Kang and Leaf, 2000*; *Kang and Leaf, 1996*). It is possible that PUFAs modulate Kv7.1/KCNE1, Nav1.5/β1, and Cav1.2/β3/α2δ channels by a similar mechanism by integrating next to the S4 voltage sensors and electrostatically attracting the voltage sensors toward their outward position. If PUFAs integrate preferentially next to the S4 that controls inactivation in Nav1.5/β1 and Cav1.2/β3/α2δ channels but next to all S4s in Kv7.1/KCNE1 channels, PUFAs would promote activation in Kv7.1/KCNE1 channels but promote inactivation in Nav1.5/β1 and Cav1.2/β3/α2δ channels. Though both PUFAs and PUFA analogues are known to modulate different ion channel activities (i.e. processes underlying activation and inactivation), it is unclear whether specific PUFAs and PUFA analogues are selective for certain ion channels or if they broadly influence the activity of several different ion channels simultaneously. PUFAs would provide an alternative to the current treatments used for LQTS. Current treatments include prescription of beta blockers or an implantable cardioverter defibrillator as means of preventing arrhythmia. However, these treatments do not target the underlying cause of LQTS, which is mutations in voltage-gated ion channels expressed in cardiomyocytes. In addition, there are some cases of LQTS where patients are resistant to current treatments (*Chockalingam et al., 2012*; *Waddell-Smith and Skinner, 2016*). PUFA analogues, however, have the potential to specifically target the ion channels that are implicated in LQTS and may benefit patients who have resistant forms of LQTS.

In this work, we characterize the channel-specific effects of different PUFAs and PUFA analogues in order to further understand which PUFAs and PUFA analogues would be the most therapeutically relevant in the treatment for different LQTS subtypes. We have found that PUFA analogues modulate Kv7.1/KCNE1, Cav1.2/β3/α2δ, and Nav1.5/β1 through different mechanisms instead of through a shared mechanism. In addition, we demonstrate that PUFA analogues exhibit a broad range of differences in selectivity for Kv7.1/KCNE1, Cav1.2/β3/α2δ, and Nav1.5/β1. PUFA analogues that are more selective for Kv7.1/KCNE1 are able to restore a prolonged ventricular action potential and prevent arrhythmia in simulated cardiomyocytes. Finally, PUFA analogues that are selective for Kv7.1/KCNE1 are able to shorten the ventricular action potential in human-induced pluripotent stem cell-derived cardiomyocytes (hiPSC-CMs).

## Results

### PUFA analogue, Lin-taurine, modulates Kv7.1/KCNE1, Cav1.2/β3/α2δ, and Nav1.5/β1 through distinct mechanisms

There are several studies supporting electrostatic activation of Kv7.1/KCNE1 channels by PUFA analogues (*Liin et al., 2015*; *Liin et al., 2016*; *Bohannon et al., 2020*; *Liin et al., 2018a*; *Larsson et al., 2018*). PUFAs are known to inhibit Nav1.5/β1 and Cav1.2/β3/α2δ channels, but there is little evidence on the mechanism of channel inhibition. Moreover, the effects of PUFA analogues on Nav1.5/β1 and Cav1.2/β3/α2δ channels remain unknown. Previous groups have suggested that PUFAs may inhibit Nav1.5/β1 and Cav1.2/β3/α2δ channels by interacting with S4 voltage sensors and stabilizing the inactivated state since there are similarities between the voltage sensor profiles of Nav1.5/β1

and Cav1.2/β3/α2δ channels. (*Kang and Leaf, 2000*; *Kang and Leaf, 1996*; *Xiao et al., 1997*; *Xiao et al., 1995*). For this reason, we hypothesize that PUFA analogues inhibit Cav1.2/β3/α2δ and Nav1.5/β1 through a shared electrostatic mechanism on S4 voltage sensors, similar to that reported with Kv7.1/KCNE1 channels. But in the case of Nav1.5/β1 and Cav1.2/β3/α2δ channels, PUFA analogues would left-shift the voltage dependence of inactivation instead of activation which is seen in Kv7.1/KCNE1. To compare the effects of different PUFA analogues on these three different channels, we here measure the currents from Kv7.1/KCNE1, Cav1.2/β3/α2δ, and Nav1.5/β1 expressed in *Xenopus* oocytes using two-electrode voltage clamp.

We first illustrate the effects of a representative PUFA analogue, Linoleoyl taurine (Lin-taurine), on the voltage dependence of activation and the conductance of Kv7.1/KCNE1 (*Figure 1A–C*). These effects are reflected in the tail current-voltage relationship where the effects on the voltage sensor are measured as a significant leftward shift in the voltage dependence of activation and the effects on the conductance are an increase in the maximal conductance upon application of Lin-taurine (*Figure 1C*). We also measure the effect of Lin-taurine on Cav1.2/β3/α2δ and Nav1.5/β1 channels (*Figures 2–3*). When we apply Lin-taurine to the Cav1.2/β3/α2δ macromolecular complex (*Figure 2A*), we see that Lin-taurine reduces current through Cav1.2/β3/α2δ (where barium (Ba$^{2+}$) is used as the charge carrier) in a dose-dependent manner (*Figure 2B–C*). However, Lin-taurine reduces Ba$^{2+}$ current without shifting the voltage-dependence of Cav1.2/β3/α2δ activation (*Figure 2B–C*). We use a depolarizing pre-pulse protocol to measure changes in voltage-dependent inactivation (*Figure 2D–E*). When we measure the effects of Lin-taurine on voltage-dependent

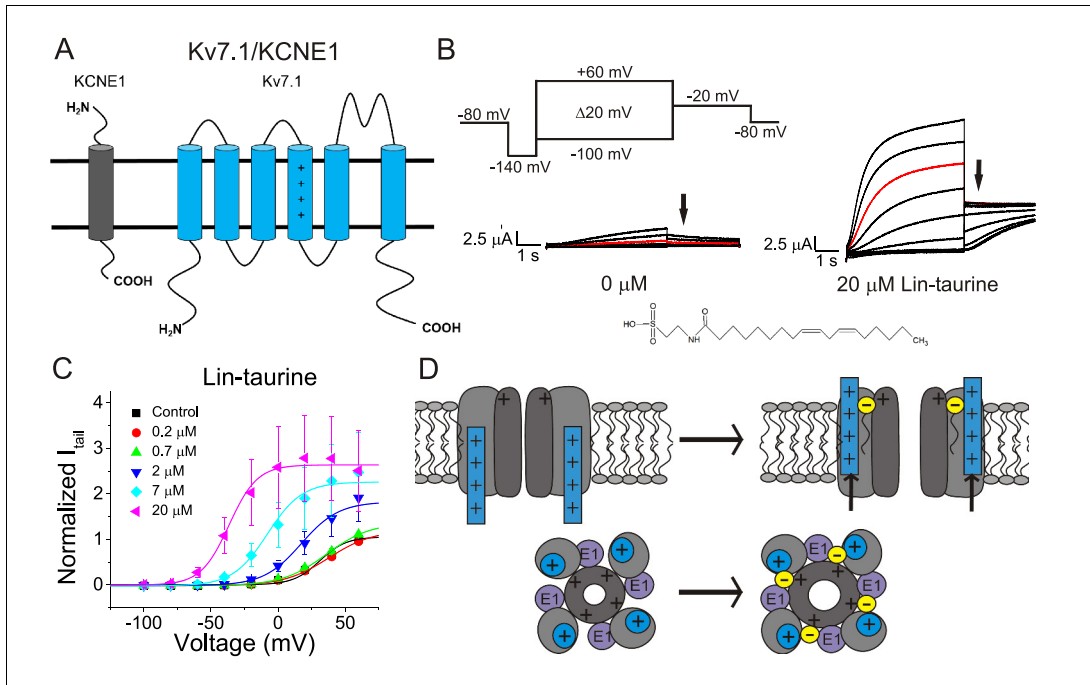

**Figure 1.** PUFA analogue, Linoleoyl-taurine, activates Kv7.1/KCNE1 channels through an electrostatic mechanism on voltage sensor and pore. (**A**) Simplified membrane topology of a single Kv7.1 α-subunit (blue) and a single KCNE1 β-subunit (grey). (**B**) Voltage protocol used to measure voltage dependence of activation and representative Kv7.1/KCNE1 current traces in control (0 µM) and 20 µM Lin-taurine. Arrows mark tail currents. (**C**) Current-voltage relationship demonstrating PUFA analogue-induced left-shift in the voltage-dependence of activation (V$_{0.5}$) and increase in maximal conductance (G$_{max}$) (mean ± SEM; n = 3). (**D**) Model in which PUFA analogues with their negatively charged head groups insert in the cell membrane close to the positively charged arginines in the voltage sensor S4 and close to a positively charged lysine in the pore. The PUFA thereby exerts an electrostatic effect on the voltage sensor to shift the voltage dependence of activation and an electrostatic effect on the pore to increase the maximum conductance (*Liin et al., 2018a*).

The online version of this article includes the following source data for figure 1:

**Source data 1.** Effects of lin-taurine on cardiac ion channels.

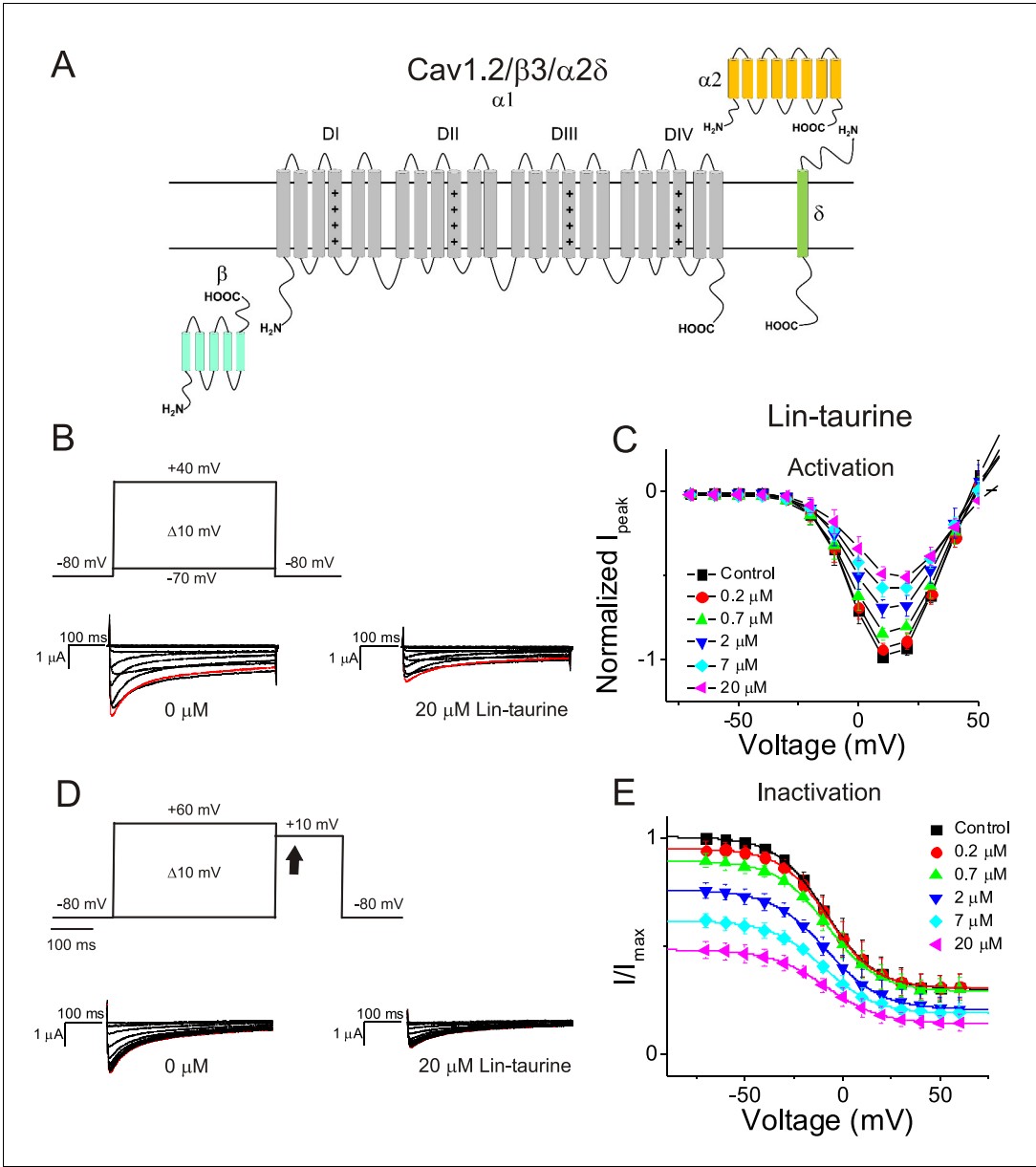

**Figure 2.** PUFA analogue, Linoleoyl-taurine, inhibits Cav1.2/β3/α2δ channels without altering channel voltage dependence. (A) Simplified membrane topology of the Cav1.2 pore-forming α-subunit (light gray) and auxiliary β- (mint) and α2δ-subunits (yellow and green). (B) Voltage protocol used to measure voltage dependence of activation and representative Cav1.2/β3/α2δ current traces in control (0 μM) and 20 μM Lin-taurine. (C) Current-voltage relationship demonstrating dose-dependent inhibition of Cav1.2/β3/α2δ currents measured from activation protocol (mean ± SEM; n = 5). (D) Voltage protocol used to measure voltage dependence of inactivation and representative Cav1.2/β3/α2δ current traces in control (0 μM) and 20 μM Lin-taurine measured at arrow. (E) Current-voltage relationship demonstrating dose-dependent inhibition of Cav1.2/β3/α2δ currents measured from inactivation protocol (mean ± SEM; n = 5). See *Figure 2—figure supplement 1* for comparison with effects on Cav1.2/β2/α2δ.

The online version of this article includes the following figure supplement(s) for figure 2:

**Figure supplement 1.** PUFA-induced effects on Cav1.2 expressed with β-subunit β2 and α2δ (Cav1.2/β2/α2δ).

inactivation, we see again a decrease in $Ba^{2+}$ currents, but surprisingly no shift in voltage-dependent inactivation (*Figure 2D–E*). This suggests that Lin-taurine does not inhibit Cav1.2/β3/α2δ channels through an electrostatic mechanism on S4 voltage sensors that shifts the voltage dependence of S4 movement, but rather through a mechanism that reduces either the number of conducting channels (potentially through an effect on the pore) or the maximum conductance of each channel. We also assessed whether the effects induced by Lin-taurine differ based on the composition of auxiliary

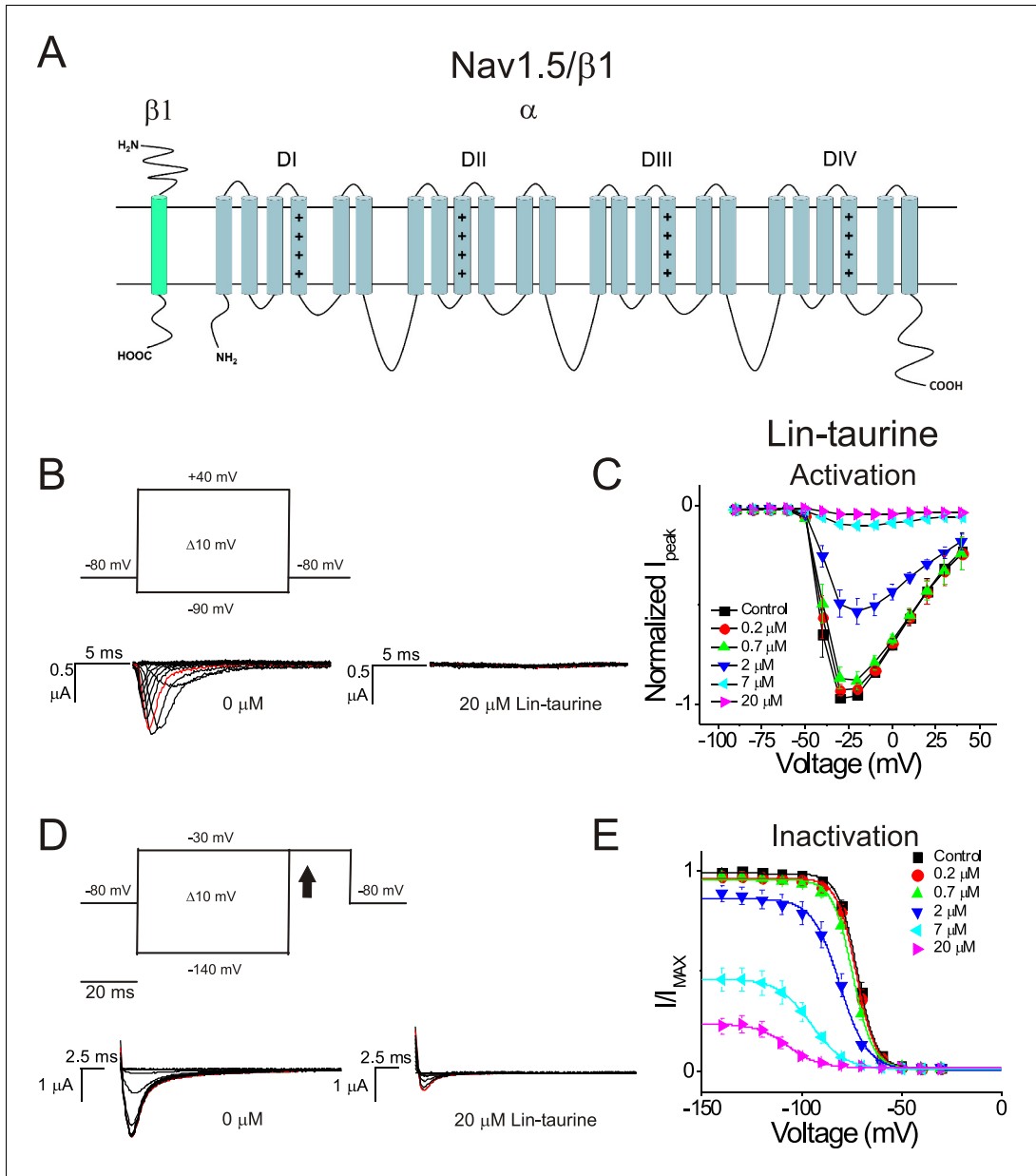

**Figure 3.** PUFA analogue, Linoleoyl-taurine, inhibits Nav1.5/β1 by shifting the voltage dependence of inactivation. (**A**) Simplified membrane topology of the Nav1.5 pore-forming α-subunit (light blue) and auxiliary β-subunit (green). (**B**) Voltage protocol used to measure voltage dependence of activation and representative Nav1.5/β1 current traces in control (0 μM) and 20 μM Lin-taurine. (**C**) Current-voltage relationship demonstrating dose-dependent inhibition of Nav1.5/β1 currents measured from activation protocol (mean ± SEM; n = 5). (**D**) Voltage protocol used to measure voltage dependence of inactivation and representative Nav1.5/β1 current traces in control (0 μM) and 20 μM Lin-taurine measured at arrow. (**E**) Current-voltage relationship demonstrating dose-dependent inhibition of Nav1.5/β1 currents and leftward shift in the voltage dependence of inactivation measured from inactivation protocol (mean ± SEM; n = 5).

subunits. Another β-subunit that can associate with Cav1.2 in cardiac tissue is the β2 subunit (*Buraei and Yang, 2013*; *Hullin et al., 2003*) and it is possible that PUFA-induced effects could be influenced by differences in channel subunit composition. However, the effects induced by Lin-taurine are not altered by co-expression with the β2 subunit (*Figure 2—figure supplement 1*). When we apply Lin-taurine to Nav1.5/β1 (*Figure 3A*) and measure voltage-dependent activation, we see a dose-dependent inhibition of $Na^+$ currents with no shift in the voltage dependence of activation (*Figure 3B–C*). However, when we measured voltage-dependent inactivation of Nav1.5/β1, we observed that Lin-taurine left-shifts the voltage dependence of inactivation (*Figure 3D–E*). In addition to the left-shift in voltage-dependent inactivation, we also observe a dose-dependent decrease in Nav1.5/β1 currents (*Figure 3E*). This suggests that Lin-taurine may also have an additional effect on the conductance of Nav1.5/β1, leading to the dose-dependent decrease in $Na^+$ currents seen on top of the leftward shift of the voltage dependence of inactivation.

Through these data, we observe that Lin-taurine modulates cardiac voltage-gated ion channels through non-identical mechanisms. Lin-taurine promotes the activation of Kv7.1/KCNE1 most likely through electrostatic effects that left-shift the voltage dependence of activation and an increase in the maximal conductance, as we have shown previously for other PUFA analogues (*Bohannon et al., 2020*; *Liin et al., 2018a*). Lin-taurine inhibits Cav1.2/β3/α2δ channels through an effect on the maximum conductance, leading to a dose-dependent decrease in $Ba^{2+}$ current but without producing any leftward shift in the voltage dependence of inactivation. In addition, Lin-taurine inhibits Nav1.5/β1 through a combination of a leftward shift in the voltage dependence of inactivation and an effect on the maximum conductance, leading to a dose-dependent decrease in $Na^+$ current. Together these findings show that Lin-taurine affects Kv7.1/KCNE1, Nav1.5/β1, and Cav1.2/β3/α2δ channels through different mechanisms.

## PUFA analogues with taurine head groups are non-selective and broadly modulate multiple cardiac ion channels, with preference for Nav1.5/β1

We have found through previous work that PUFA analogues with taurine head groups are good activators of the Kv7.1/KCNE1 channels due to the low pKa of the taurine head group (*Liin et al., 2015*; *Liin et al., 2016*; *Bohannon et al., 2020*). Having a lower pKa allows the taurine head group to be fully negatively charged at physiological pH so that it has maximal electrostatic effects on Kv7.1/KCNE1 channels (*Liin et al., 2016*; *Bohannon et al., 2020*). We tested a set of PUFA analogues with taurine head groups on Kv7.1/KCNE1, Cav1.2/β3/α2δ, and Nav1.5/β1 channels to determine if these effects are selective for the Kv7.1/KCNE1 channel or if other taurine analogues also modulate Cav1.2/β3/α2δ and Nav1.5/β1 channels.

We demonstrated in recent work that Lin-taurine, N-AT, Pin-taurine, and DHA-taurine (*Figure 4*, top) promote the activation of the Kv7.1/KCNE1 channel, by left-shifting the voltage-dependence of activation ($\Delta V_{0.5}$) and increasing the maximal conductance of the Kv7.1/KCNE1 channel ($G_{max}/G_{max0}$) (*Liin et al., 2016*; *Bohannon et al., 2020*). At 7 µM, the $\Delta V_{0.5}$ is $-39.9 \pm 3.6$ mV for Lin-taurine (p=0.008), $-1.8 \pm 2.6$ mV for N-AT (p=0.5, ns), $-23.8 \pm 2.7$ mV for Pin-taurine (p=0.003), and $-45.3 \pm 2.9$ mV for DHA-taurine (p=0.004) (mean ± SEM, *Figure 4A* and *Figure 4—figure supplement 1*). At 7 µM, the $G_{max}/G_{max0}$ is $1.9 \pm 0.6$ for Lin-taurine (p=0.001), $0.9 \pm 0.03$ for N-AT (p=0.98, ns), $2.3 \pm 0.4$ for Pin-taurine (p=0.001), and $1.7 \pm 0.1$ for DHA-taurine (p=0.001) (mean ± SEM, *Figure 4B* and *Figure 4—figure supplement 1*). N-arachidonoyl taurine (N-AT) has been shown to activate Kv7.1/KCNE1 by left-shifting the voltage-dependence and increasing the $G_{max}$ at 70 µM (*Liin et al., 2016*). However, the effects of N-AT at concentrations lower that 70 µM have not yet been explored. Here, we used lower concentrations (0.2, 0.7, 2, 7, and 20 µM) with the goal of understanding the selectivity of N-AT for cardiac ion channels at lower concentrations.

The four taurine compounds inhibit Cav1.2/β3/α2δ current in a dose-dependent manner without shifting the voltage dependence of activation (*Figure 4—figure supplement 2*) or the voltage dependence of inactivation ($\Delta V_{0.5}$, *Figure 4C* and *Figure 4—figure supplements 2* and *4*). Instead, the four taurine compounds decrease $G_{max}$ ($G_{max}/G_{max0}$, *Figure 4D*). At 7 µM, the $\Delta V_{0.5}$ is $-1.9 \pm 2.5$ mV for Lin-taurine (p=0.49, ns), $0.6 \pm 2.0$ mV for N-AT (p=0.8, ns), $4.1 \pm 2.2$ mV for Pin-taurine (p=0.13, ns), and $0.2 \pm 0.8$ mV for DHA-taurine (p=0.85, ns) (mean ± SEM, *Figure 4C*, *Figure 4—figure supplements 2* and *4*). At 7 µM, the $G_{max}/G_{max0}$ is $0.6 \pm 0.05$ for Lin-taurine

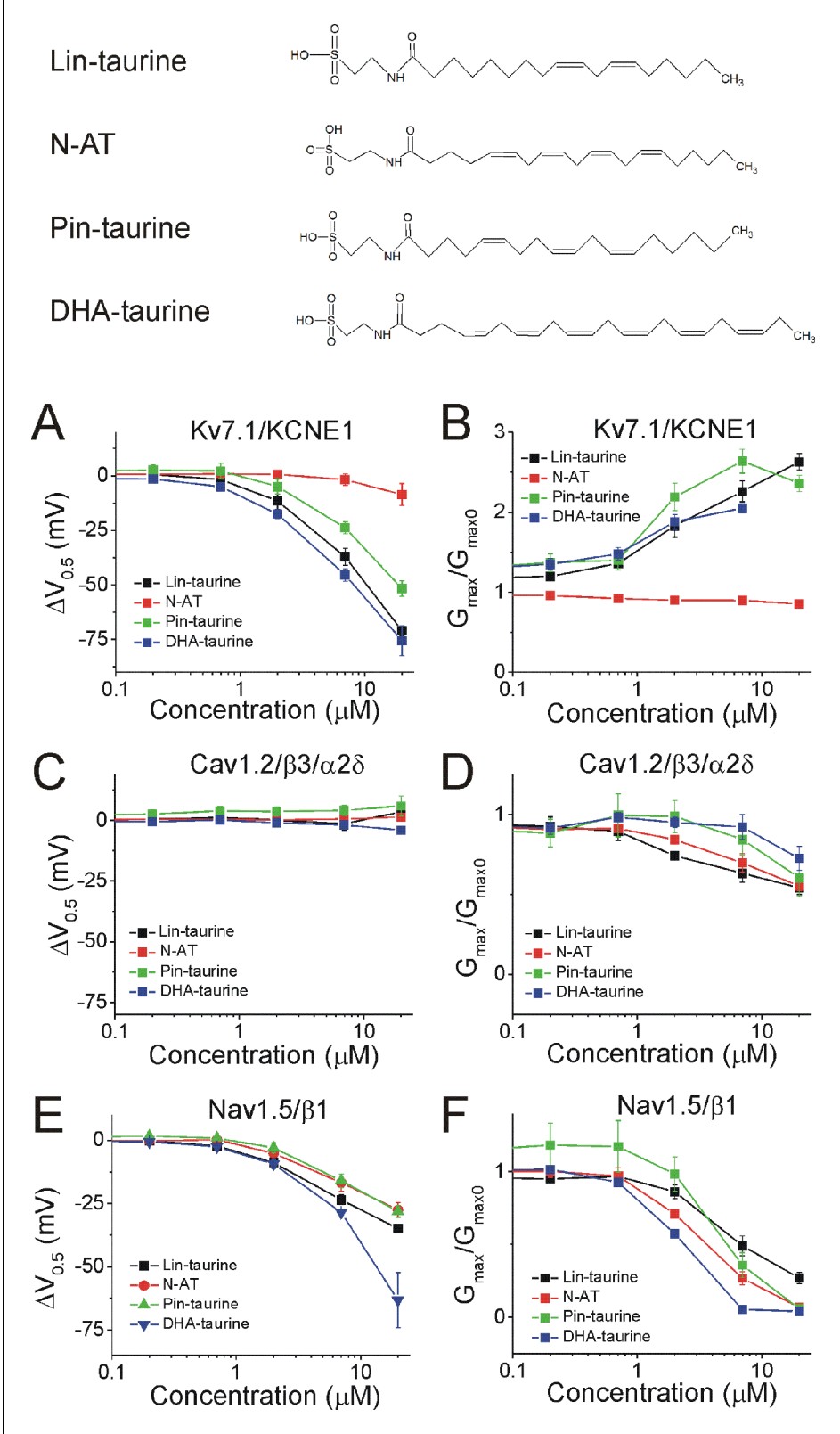

**Figure 4.** PUFA analogues with taurine head groups have the tendency to broadly modulate Kv7.1/KCNE1, Cav1.2/β3/α2δ, and Nav1.5/β1 channels. Structures for PUFA analogues with taurine head groups: Lin-taurine, N-AT, Pin-taurine, and DHA-taurine (top). (A–B) Dose dependent shifts in the (A) voltage dependence of activation ($\Delta V_{0.5}$) and (B) changes in maximal conductance ($G_{max}$) for Kv7.1/KCNE1 induced by Lin-taurine (black squares;

*Figure 4 continued on next page*

*Figure 4 continued*

n = 3; mean ± SEM), N-AT (red circles; n = 5; mean ± SEM), Pin-taurine (green triangles; n = 4; mean ± SEM), and DHA-taurine (blue triangles; n = 3; mean ± SEM). (**C–D**) Dose dependent shifts in the (**C**) voltage dependence of inactivation ($\Delta V_{0.5}$) and (**D**) changes in maximal conductance ($G_{max}$) for Cav1.2/β3/α2δ induced by Lin-taurine (black squares; n = 3; mean ± SEM), N-AT (red circles; n = 3; mean ± SEM), Pin-taurine (green triangles; n = 5; mean ± SEM), and DHA-taurine (blue triangles; n = 3; mean ± SEM). (**E–F**) Dose dependent shifts in the (**E**) voltage dependence of inactivation ($\Delta V_{0.5}$) and (**F**) changes in maximal conductance ($G_{max}$) for Nav1.5/β1 induced by Lin-taurine (black squares; n = 3; mean ± SEM), N-AT (red circles; n = 3; mean ± SEM), Pin-taurine (green triangles; n = 4; mean ± SEM), and DHA-taurine (blue triangles; n = 3; mean ± SEM). See *Figure 4—figure supplements 1– 6* for more details.

The online version of this article includes the following source data and figure supplement(s) for figure 4:

**Source data 1.** Effects of N-AT on cardiac ion channels.
**Source data 2.** Effects of pin-taurine on cardiac ion channels.
**Source data 3.** Effects of DHA-taurine on cardiac ion channels.
**Figure supplement 1.** Raw current traces for PUFA analogues on Kv7.1/KCNE1.
**Figure supplement 2.** Raw current traces for PUFA analogues on Cav1.2/β3/α2δ.
**Figure supplement 3.** PUFA-induced changes in $I/I_0$ normalized by concentration show no changes in voltage-dependent activation of Cav1.2/β3/α2δ and Nav1.5/β1 channels.
**Figure supplement 4.** Internally normalized steady state inactivation curves for Cav1.2/β3/α2δ.
**Figure supplement 5.** Raw current traces for PUFA analogues on Nav1.5/β1 channels.
**Figure supplement 6.** Internally normalized steady state inactivation curves for Nav1.5/β1.

---

(p=0.001), 0.7 ± 0.1 for N-AT (p=0.03), 0.8 ± 0.1 for Pin-taurine (p=0.2, ns), and 1.1 ± 0.1 for DHA-taurine (p=0.38, ns) (mean ± SEM, *Figure 4D* and *Figure 4—figure supplement 2*).

Lastly, the four taurine compounds inhibit Nav1.5/β1 current without shifting the voltage dependence of activation (*Figure 4—figure supplements 3* and *5*), but by left-shifting the voltage dependence of inactivation ($\Delta V_{0.5}$, *Figure 4E* and *Figure 4—figure supplements 5–6*) and decreasing $G_{max}$ ($G_{max}/G_{max0}$, *Figure 4F* and *Figure 4—figure supplement 5*). At 7 μM, the $\Delta V_{0.5}$ is −23.5 ± 1.9 mV for Lin-taurine (p=0.001), −16.7 ± 3.5 mV for N-AT (p=0.04), −16 ± 2.7 mV for Pin-taurine (p=0.01), and −28.5 ± 0.6 mV for DHA-taurine (p=0.005) (mean ± SEM, *Figure 4E*, *Figure 4— figure supplements 5–6*). At 7 μM, the $G_{max}/G_{max0}$ is 0.5 ± 0.07 for Lin-taurine (p=0.005), 0.3 ± 0.04 for N-AT (p=0.004), 0.4 ± 0.09 for Pin-taurine (p=0.005), and 0.05 ± 0.01 for DHA-taurine (p<0.001) (mean ± SEM, *Figure 4F* and *Figure 4—figure supplement 5*). These results together suggest that PUFA analogues with taurine head groups exhibit broad selectivity for multiple ion channels.

## PUFA analogues with glycine head groups tend to be more selective for Kv7.1/KCNE1 with lower affinity for Cav1.2/β3/α2δ and Nav1.5/β1

PUFA analogues with glycine head groups (namely DHA-glycine) have also been shown to effectively activate the Kv7.1/KCNE1 channel (*Liin et al., 2015*; *Bohannon et al., 2020*). The glycine head group has a lower pKa than regular PUFAs with a carboxyl head group (*Liin et al., 2015*; *Bohannon et al., 2020*), thereby allowing the head group to be more deprotonated and partially negatively charged at physiological pH. For this reason, PUFA analogues with glycine head groups are able to electrostatically activate Kv7.1/KCNE1 channels (*Liin et al., 2015*). We here tested several glycine compounds on Kv7.1/KCNE1, Cav1.2/β3/α2δ, and Nav1.5/β1 channels to determine whether they have selective or non-selective effects on cardiac ion channels.

We demonstrated in recent work that Lin-glycine, Pin-glycine, and DHA-glycine (*Figure 5*, top) promote the activation of the Kv7.1/KCNE1 channel, by left-shifting the voltage-dependence of activation ($\Delta V_{0.5}$) and increasing the maximal conductance of the Kv7.1/KCNE1 channel ($G_{max}/G_{max0}$) (*Liin et al., 2016*; *Bohannon et al., 2020*). At 7 μM, the $\Delta V_{0.5}$ is −23.8 ± 1.6 mV for Lin-glycine (p<0.001), −8.7 ± 1.8 mV for Pin-glycine (p=0.04), and −10.5 ± 1.0 mV for DHA-glycine (p=0.002) (mean ± SEM, *Figure 5A* and *Figure 4—figure supplement 1*). At 7 μM, the $G_{max}/G_{max0}$ is 2.3 ± 0.2 for Lin-glycine (p=0.008), 1.7 ± 0.1 for Pin-glycine (p=0.03), and 1.9 ± 0.2 for DHA-glycine (p=0.03) (mean ± SEM, *Figure 5B* and *Figure 4—figure supplement 1*).

The three glycine compounds do not inhibit Cav1.2/β3/α2δ current (*Figure 5C–D*). At 7 μM, the $\Delta V_{0.5}$ is −3.0 ± 1.6 mV for Lin-glycine (p=0.2, ns), −3.1 ± 4.2 mV for Pin-glycine (p=0.54, ns), and

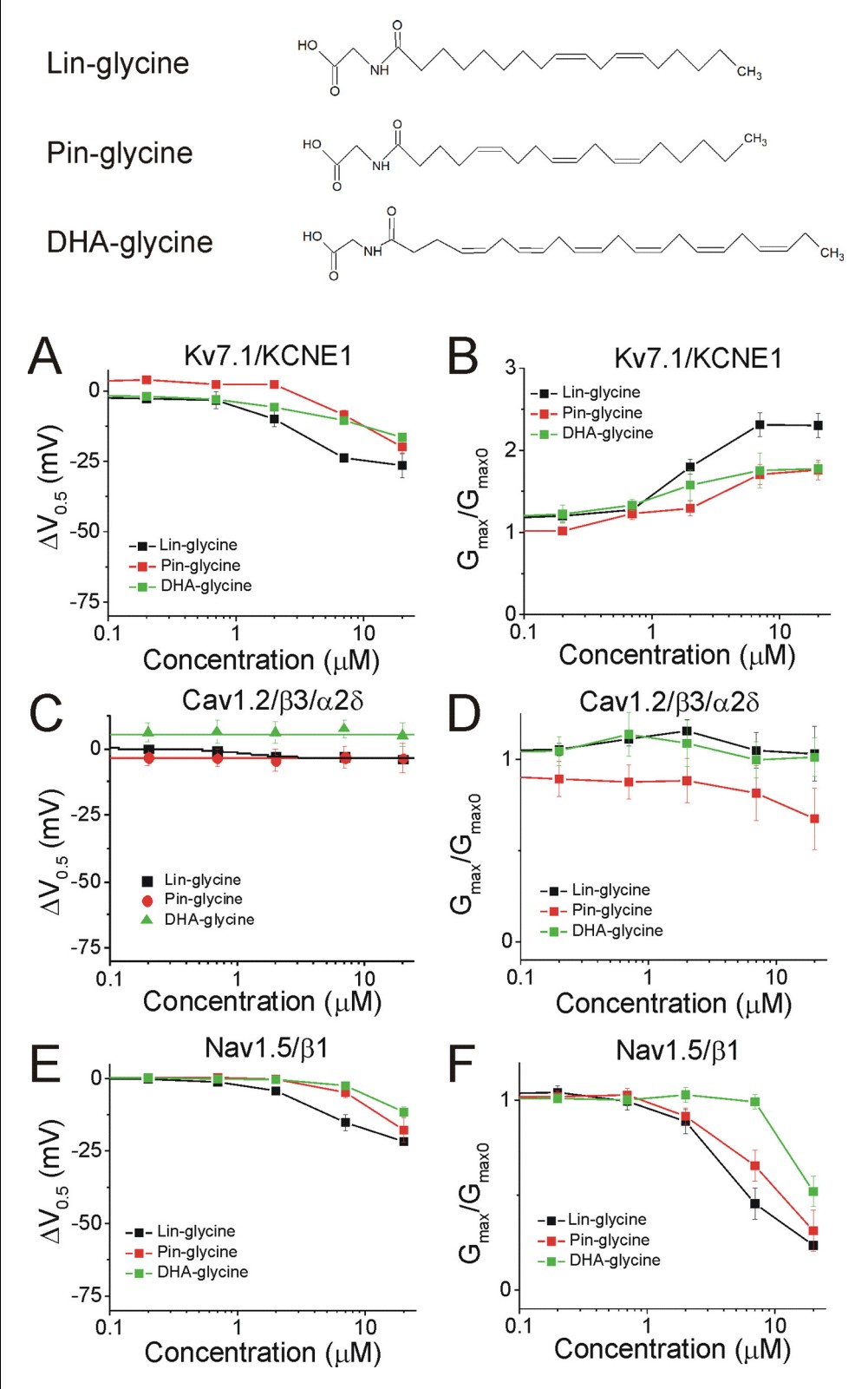

**Figure 5.** PUFA analogues with glycine head groups tend to be more selective for Kv7.1/KCNE1 than Cav1.2/β3/α2δ and Nav1.5/β1 channels. Structures for PUFA analogues with glycine head group: Lin-glycine, Pin-glycine, and DHA-glycine (top). (A–B) Dose dependent shifts in the (A) voltage dependence of activation ($\Delta V_{0.5}$) and (B) changes in maximal conductance ($G_{max}$) for Kv7.1/KCNE1 induced by Lin-glycine (black squares; n = 4;

*Figure 5 continued on next page*

*Figure 5 continued*

mean ± SEM), Pin-glycine (red circles; n = 3; mean ± SEM), and DHA-glycine (green triangles; n = 4; mean ± SEM). (C–D) Dose dependent shifts in the (C) voltage dependence of inactivation ($\Delta V_{0.5}$) and (D) changes in maximal conductance ($G_{max}$) for Cav1.2/β3/α2δ induced by Lin-glycine (black squares; n = 4; mean ± SEM), Pin-glycine (red circles; n = 3; mean ± SEM), and DHA-glycine (green triangles; n = 3; mean ± SEM). (E–F) Dose dependent shifts in the (E) voltage dependence of inactivation ($\Delta V_{0.5}$) and (F) changes in maximal conductance ($G_{max}$) for Nav1.5/β1 induced by Lin-glycine (black squares; n = 4; mean ± SEM), Pin-glycine (red circles; n = 4; mean ± SEM), and DHA-glycine (green triangles; n = 7; mean ± SEM). See *Figure 4—figure supplements 1–6* for more details.

The online version of this article includes the following source data for figure 5:

**Source data 1.** Effects of lin-glycine on cardiac ion channels.
**Source data 2.** Effects of pin-glycine on cardiac ion channels.
**Source data 3.** Effects of DHA-glycine on cardiac ion channels.

---

7.6 ± 3.1 mV for DHA-glycine (p=0.13, ns) (mean ± SEM, *Figure 5C*, *Figure 4—figure supplements 2* and *4*). At 7 µM, the $G_{max}/G_{max0}$ is 1.0 ± 0.1 for Lin-glycine (p=0.7, ns), 0.8 ± 0.1 for Pin-glycine (p=0.34, ns), and 1.0 ± 0.1 for DHA-glycine (p=0.98, ns) (mean ± SEM, *Figure 5D* and *Figure 4—figure supplement 2*).

Lastly, the three glycine compounds inhibit Nav1.5/β1 current without shifting the voltage dependence of activation (*Figure 4—figure supplements 3* and *5*), but by left-shifting the voltage dependence of inactivation ($\Delta V_{0.5}$, *Figure 5E* and *Figure 4—figure supplements 5–6*) and also by decreasing the $G_{max}$ ($G_{max}/G_{max0}$, *Figure 5F* and *Figure 4—figure supplement 5*). At 7 µM, the $\Delta V_{0.5}$ is −15.2 ± 2.8 mV for Lin-glycine (p=0.01), −4.7 ± 1.9 mV for Pin-glycine (p=0.09, ns), and −2.4 ± 0.6 mV for DHA-glycine (p=0.01) (mean ± SEM, *Figure 5E*, *Figure 4—figure supplements 5–6*). At 7 µM, the $G_{max}/G_{max0}$ is 0.5 ± 0.1 for Lin-glycine (p=0.007), 0.7 ± 0.08 for Pin-glycine (p=0.03), and 1.0 ± 0.04 for DHA-glycine (p=0.84, ns) (mean ± SEM, *Figure 5F* and *Figure 4—figure supplement 5*). These results suggest that PUFA analogues with glycine head groups tend to be more selective for the cardiac Kv7.1/KCNE1 channel and tend to have no effect or only effects at higher concentrations for Cav1.2/β3/α2δ and Nav1.5/β1 channels.

## PUFA analogues with glycine head groups are more selective for $I_{Ks}$ currents

We found that the PUFA analogues investigated here have several different effects on the same channel (e.g. they alter the voltage dependence and the conductance at the same time). To evaluate the total effects of PUFA analogues on channel currents, we compared the dose response curves for $I_{Ks}$ (Kv7.1/KCNE1), $I_{CaL}$ (Cav1.2/β3/α2δ), and $I_{NaV}$ (Nav1.5/β1) at 0 mV ($I/I_0$) for each PUFA analogue (*Figure 6*). The four taurine compounds inhibit $I_{NaV}$ to almost completion in the range of doses tested and in general with higher affinity than their effects on $I_{Ks}$ (*Figure 6A–D*). The exception is DHA-taurine that has similar affinity for $I_{Na}$ and $I_{Ks}$ (*Figure 6C*). The four taurine compounds activate $I_{Ks}$ significantly in the range of doses tested (*Figure 6A–D*) expect for N-AT for which the effects on $I_{Ks}$ appears at concentrations > 20 µM (*Liin et al., 2016*; *Figure 6D*). The four taurine compounds inhibit $I_{CaL}$ in the range of doses tested (inhibition by Pin-taurine is not clearly dose dependent and some of this reduction in current could possibly be due to rundown) but to a lesser extent than their inhibition of $I_{Na}$ (*Figure 6A–D*). In contrast, the three glycine compounds do not inhibit $I_{CaL}$ in the range of doses tested (Pin-glycine show some inhibition, but it is not clearly dose dependent and some of this reduction in current could possibly be due to rundown) (*Figure 6E–F*). Instead, the three glycine compounds inhibit $I_{NaV}$, but in general with a lower affinity than the corresponding taurine compounds (e.g. compare Lin-taurine versus Lin-glycine) (*Figure 6E–F*). The three glycine compounds activate $I_{Ks}$ to a lesser extent, but with a similar affinity, than the taurine compounds (expect N-AT) (*Figure 6E–F*). Overall, when we compare the effects of different PUFA analogues on $I_{Ks}$, $I_{NaV}$, and $I_{CaL}$ at a specific concentration (e.g. 7 µM; *Table 1*), we find that there are some PUFA analogues that affect all three channels (e.g. Lin-taurine and Pin-taurine) whereas there are other PUFA analogues that are more selective for one of the three channels in the concentration range tested (e.g. DHA-glycine). In general, PUFA analogues with taurine head groups are more broadly selective whereas PUFA analogues with glycine head groups are more selective for $I_{Ks}$ currents.

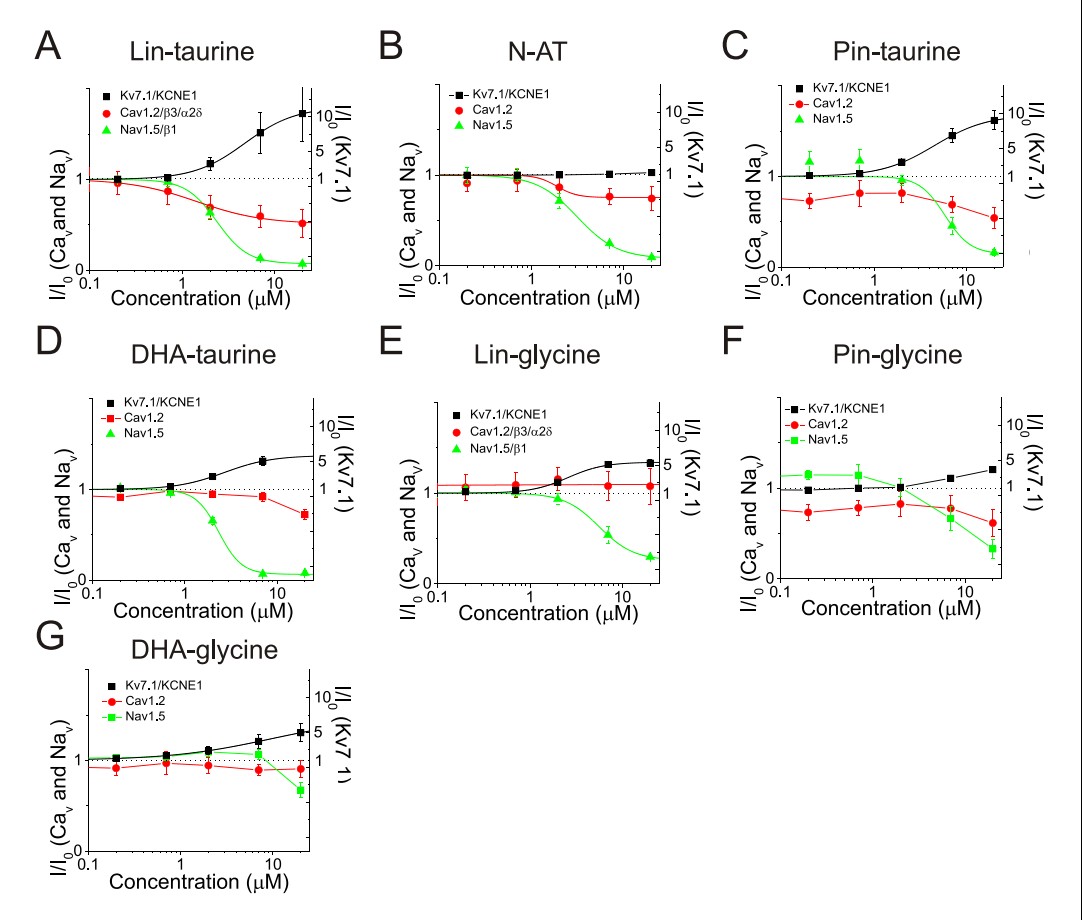

**Figure 6.** Dose response curves for PUFAs on $I_{Ks}$, $I_{CaL}$, and $I_{NaV}$ at 0 mV. Dose response of $I_{Ks}$, $I_{CaL}$, and $I_{NaV}$ currents ($I/I_0$) at 0 mV for (**A**) Lin-taurine (Km of $I_{Ks}$ = 11.4 ± 0.4 μM; Km of $I_{CaL}$ = 1.4 ± 0.4 μM; Km of $I_{NaV}$ = 2.4 ± 0.04 μM; mean ± SEM), (**B**) N-AT (Km of $I_{Ks}$ = NA; Km of $I_{CaL}$ = 2.5 ± 1.9 μM; Km of $I_{NaV}$ = 3.1 ± 0.3 μM; mean ± SEM), (**C**) Pin-taurine (Km of $I_{Ks}$ = 4.5 ± 0.2 μM; Km of $I_{CaL}$ = NA; Km of $I_{NaV}$ = 5.8 ± 1.7 μM; mean ± SEM), (**D**) DHA-taurine (Km of $I_{Ks}$ = 5.9 ± 0.3 μM; Km of $I_{CaL}$ >20 μM; Km of $I_{NaV}$ = 2.3 ± 0.1 μM; mean ± SEM), (**E**) Lin-glycine (Km of $I_{Ks}$ = 5.4 ± 0.2 μM; Km of $I_{CaL}$ = NA; Km of $I_{NaV}$ = 5.6 ± 0.5 μM; mean ± SEM), (**F**) Pin-glycine (Km of $I_{Ks}$ > 20 μM; Km of $I_{CaL}$ = NA; Km of $I_{NaV}$ >20 μM; mean ± SEM), and (**G**) DHA-glycine (Km of $I_{Ks}$ > 20 μM; Km of $I_{CaL}$ = NA; Km of $I_{NaV}$ >20 μM; mean ± SEM),.

## Selective Kv7.1/KCNE1 channel activators have antiarrhythmic effects in simulated cardiomyocytes

We next tried to understand what kind of compound is the most effective at shortening the action potential duration. To determine whether selective PUFA analogues or non-selective PUFA analogues can shorten the action potential duration, we simulated the effects of applying the PUFA analogues on human cardiomyocyte using the O'Hara-Rudy dynamic (ORd) model (*O'Hara et al., 2011*) by modifying parameters for the voltage dependence and conductance for individual channels to reflect our experimental PUFA-induced effects. We simulated the effects of PUFA analogues that are non-selective modulators for cardiac ion channels (i.e. N-AT, Lin-taurine, Pin-taurine, DHA-taurine, and Lin-glycine) at concentrations of 0.7 μM, 2 μM, and 7 μM (*Figure 7A–E*). In most cases, we saw little change in the ventricular action potential until we simulated the effects of these PUFA analogues at 7 μM, where we were unable to elicit an action potential (*Figure 7A–E*). One exception was the effect of applying DHA-taurine, in which case we observed a small shortening of the action potential at 0.7 μM, but then an abnormal action potential upstroke and prolongation of the action potential at 2 μM (*Figure 7D*). This is likely due to the potent block of Nav1.5/β1 channels, causing the action potential to be largely calcium dependent. But again, at 7 μM DHA-taurine, we were unable to elicit an action potential (*Figure 7D*). However, the PUFA analogues that were more selective for Kv7.1/KCNE1, such as Pin-glycine and DHA-glycine (at 7 μM) induce a slight shortening of

**Table 1.** Summary of PUFA effects on current and apparent affinity.

$I/I_0$ represents the relative current of the specified channel. The $K_m$ indicates the concentration at which half the maximal effect on $I/I_0$ occurs and is used as a measure of the apparent affinity of the PUFA analogue. Data is represented as mean ± SEM. Comparisons were made using One-way ANOVA and Student's t-test. Significance level is set to p=0.05.

| PUFA Analogue | $I_{Ks}$ $I/I_0$ (7 µM) (mean ± SEM) | $K_m$ $I_{Ks}$ (µM) (mean ± SEM) | Adj $R^2$ of Hill Fit | $I_{Ca}$ $I/I_0$ (7 µM) (mean ± SEM) | $K_m$ $I_{Ca}$ (µM) (mean ± SEM) | Adj $R^2$ of Hill Fit | $I_{Na}$ $I/I_0$ (7 µM) (mean ± SEM) | $K_m$ $I_{Na}$ (µM) (mean ± SEM) | Adj $R^2$ of Hill Fit |
|---|---|---|---|---|---|---|---|---|---|
| Lin-taurine | 7.7 ± 2.9 (p=0.14) | 11.4 ± 0.4 | 0.99 | 0.6 ± 0.1 (p=0.01) | 1.4 ± 0.4 | 0.98 | 0.1 ± 0.02 (p=0.0001) | 2.4 ± 0.04 | 0.99 |
| N-AT | 1.12 ± 0.1 (p=0.27) | NA | NA | 0.8 ± 0.1 (p=0.1) | 2.5 ± 1.9 | 0.70 | 0.2 ± 0.01 (p=0.001) | 3.1 ± 0.3 | 0.99 |
| Pin-taurine | 6.8 ± 1.0 (p=0.01) | 4.5 ± 0.2 | 0.99 | 0.7 ± 0.1 (p=0.02) | NA | NA | 0.5 ± 0.1 (p=0.01) | 5.8 ± 1.7 | 0.89 |
| DHA-taurine | 5.1 ± 0.7 (p=0.03) | 5.9 ± 0.3 | 0.99 | 0.9 ± 0.1 (p=0.23) | >20 | 0.75 | 0.07 ± 0.01 (p=0.0001) | 2.3 ± 0.1 | 0.99 |
| Lin-glycine | 5.1 ± 0.4 (p=0.002) | 5.4 ± 0.2 | 0.99 | 1.1 ± 0.2 (p=0.4) | NA | NA | 0.5 ± 0.1 (p=0.02) | 5.6 ± 0.5 | 0.96 |
| Pin-glycine | 2.5 ± 0.2 (p=0.02) | >20 | 0.98 | 0.8 ± 0.1 (p=0.26) | NA | NA | 0.7 ± 0.1 (p=0.09) | >20 | 0.90 |
| DHA-glycine | 3.7 ± 1.0 (p=0.07) | >20 | 0.99 | 0.9 ± 0.1 (p=0.19) | NA | NA | 1.1 ± 0.05 (p=0.24) | >20 | 1 |

the wild type ventricular action potential (*Figure 7F–G*). For Pin-glycine and DHA-glycine, we induced Long QT Type 2 by simulating 25% block of the hERG channel, which generates the rapid component of the delated rectifier potassium current ($I_{Kr}$). 25% hERG block prolongs the ventricular action potential by 50 ms (*Figure 7F–G*). Application of Pin-glycine or DHA-glycine at 7 µM in the simulation partially restores the duration of the ventricular action potential (*Figure 7F–G*). The percent shortening of the APD90 by DHA-glycine was slightly larger for lower (40 beats-per-minute) and higher (200 bpm) simulations of the heart rate than with a heart rate with 60 bpm (40 bmp: −4.3%; 60 bpm: −3.5%; 200bpm: −4.5%; *Figure 7—figure supplement 1*). At high bpm, the $I_{Ks}$ current increases due to a rate-dependent accumulation of open Kv7.1/KCNE1 channels (*O'Hara et al., 2011*; *Figure 7—figure supplement 2*), which most likely underlies the larger percent shortening of the APD90 at 200 bpm. The reason for the increased percent shortening at 40 bpm compared to 60 bpm is not clear (*Figure 7—figure supplement 2*). In addition to simulating the effects of PUFA analogues on the ventricular action potential duration, we also simulated the ability of 7 µM DHA-glycine (the most selective Kv7.1/KCNE1 activator) to prevent arrhythmia by simulating early afterdepolarizations using 0.1 µM dofetilide. Dofetilide is a blocker of the hERG channel and increases the susceptibility for early afterdepolarizations (*O'Hara et al., 2011*; *Figure 7H*). When we simulate 0.1 µM dofetilide + 7 µM DHA-glycine, we are able to suppress early afterdepolarizations (*Figure 7H*).

## DHA-glycine shortens the ventricular action potential in human induced pluripotent stem cell-derived cardiomyocytes

To analyze the effects of DHA-glycine on the human cardiac action potential, we optically recorded the calcium transients (CaTs) of human induced pluripotent stem cell-derived cardiomyocytes (hiPSC-CM) grown in a monolayer (*Spencer et al., 2014*). The CaTs were recorded in the absence and in the presence of 30 µM DHA-glycine (*Figure 8*). We use the CaT duration as an estimate of the action potential duration (APD) and measured the CaT duration at 90% of return to baseline to obtain an estimate of the CaT APD90 value. Due to spontaneous beating of hiPSC-derived cardiomyocytes, the APD90 values of the CaTs were corrected by frequency in each experiment using the modified Fridericia's formula (APD90c) (*Fridericia, 2003*). The CaT APD90c obtained from hiPSC-CM monolayers in control conditions was 472 ± 27 ms (n = 3), whereas, in the presence of 30 µM of DHA-glycine, the APD90c observed was 317 ± 9 ms (n = 3) (*Figure 8A–B*). These results show that DHA-glycine shortens the duration of the calcium transients in these cells, as if DHA-glycine shortens the cardiac action potential duration of hiPSC-derived cardiomyocytes.

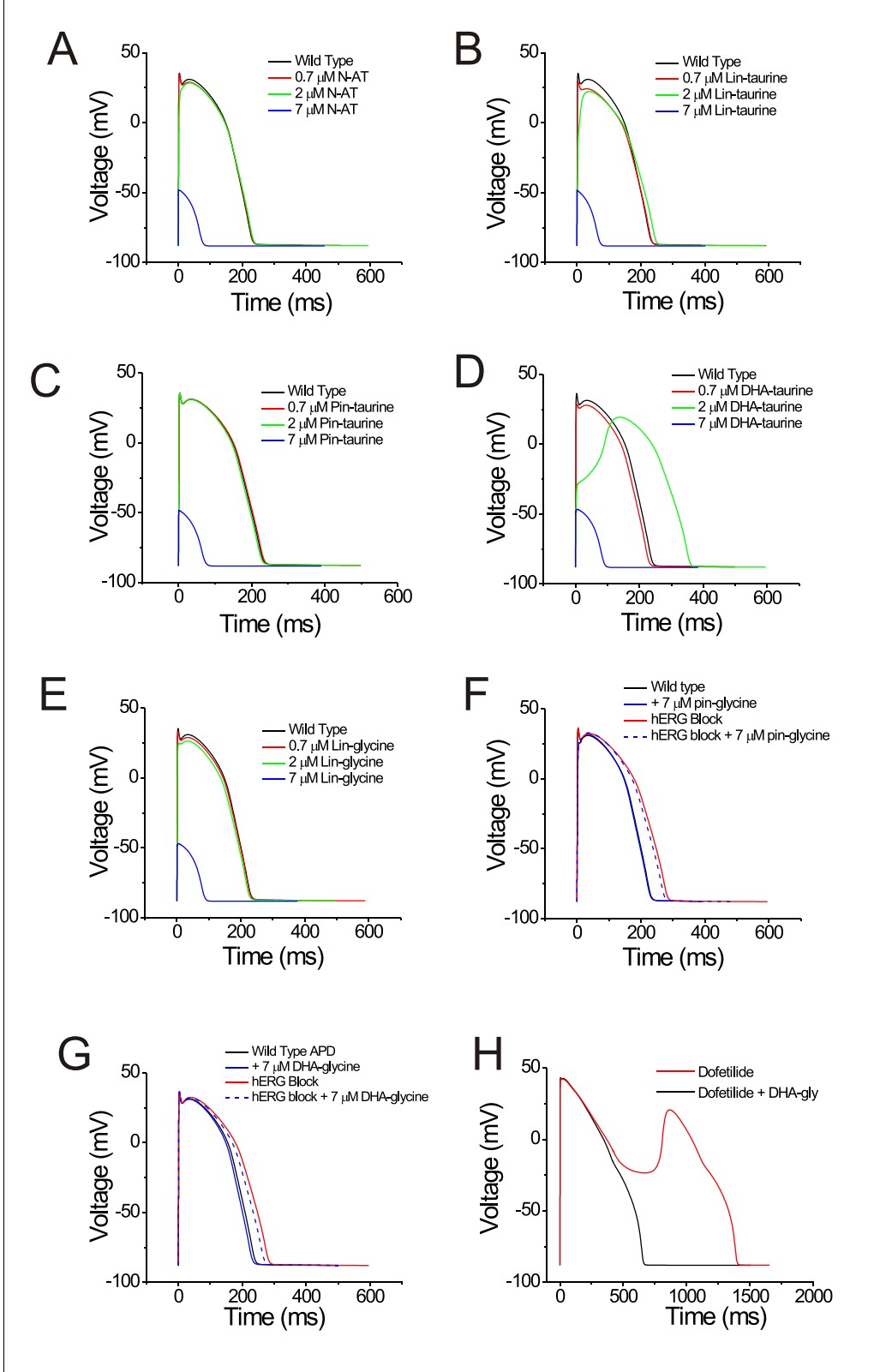

**Figure 7.** PUFAs that are selective for Kv7.1/KCNE1 channels partially restore prolonged ventricular action potential and suppress early afterdepolarizations in cardiomyocyte simulations. (A–G) Simulated ventricular action potential in wild type cardiomyocytes (black) and in the presence of (A) 0.7 (red), 2 (green), and 7 µM N-AT (blue), (B) 0.7 (red), 2 (green), and 7 µM lin-taurine (blue), (C) 0.7 (red), 2 (green), and 7 µM pin-taurine (blue), (D) 0.7 (red), 2 (green), and 7 µM DHA-taurine (blue), (E) 0.7 (red), 2 (green), and 7 µM lin-glycine (blue), (F) 7 µM pin-glycine (blue solid), following 25%

*Figure 7 continued on next page*

*Figure 7 continued*

hERG block (red) and in the presence of 7 µM pin-glycine under 25% hERG block (blue dashed), and **(G)** 7 µM DHA-glycine (blue solid), following 25% hERG block (red) and in the presence of 7 µM DHA-glycine under 25% hERG block (blue dashed). **(H)** Early afterdepolarizations induced by dofetilide application (red) and suppression of early afterdepolarizations by 7 µM DHA-glycine in the presence of dofetilide (black). See *Figure 7—figure supplement 1* for rate dependence of the effects.

The online version of this article includes the following figure supplement(s) for figure 7:

**Figure supplement 1.** Rate dependence of the simulated ventricular action potential and effects of DHA-glycine.

**Figure supplement 2.** Rate dependence of the $I_{Ks}$ current during simulated ventricular action potentials and effects of DHA-glycine.

## Discussion

We show here that PUFA analogues have different mechanisms of action on Kv7.1/KCNE1, Cav1.2/β3/α2δ, and Nav1.5/β1 channels. We have previously shown that PUFAs promote the activation of Kv7.1/KCNE1 channels through the lipoelectric mechanism where the negatively charged PUFA head group electrostatically attracts both the S4 voltage sensor (facilitating its upward movement and channel opening) and K326 in S6 (increasing the maximal conductance) (*Börjesson et al., 2008*; *Liin et al., 2018a*; *Börjesson and Elinder, 2011*). In both Cav1.2/β3/α2δ and Nav1.5/β1 channels, PUFAs and PUFA analogues inhibit channel currents. We have found that PUFA analogues cause a dose-dependent reduction in the currents through Cav1.2/β3/α2δ channels. Surprisingly this inhibition occurs with no effect on the voltage dependence of either activation or inactivation. In Nav1.5/β1 channels, PUFA analogues cause inhibition through a dose-dependent decrease in currents, with both a left-shifting effect on the voltage dependence of inactivation and a decrease in conductance. We also demonstrate that PUFA analogues vary in their selectivity for voltage-gated ion channels. The selectivity depends on the specific concentration of PUFA analogues applied, because several compounds have non-overlapping dose response curves for their effects on the three different channels (*Figure 6*). We also found that PUFA analogues with taurine head groups tend to have broad modulatory effects on Kv7.1/KCNE1, Nav1.5/β1 and Cav1.2/β3/α2δ channels, generally displaying higher apparent affinity for Cav1.2/β3/α2δ and Nav1.5/β1 channels. Conversely, PUFA analogues with glycine head groups tend to be more selective for Kv7.1/KCNE1 channels and display lower apparent affinity for Cav1.2/β3/α2δ and Nav1.5/β1 channels. By understanding the effects of PUFA

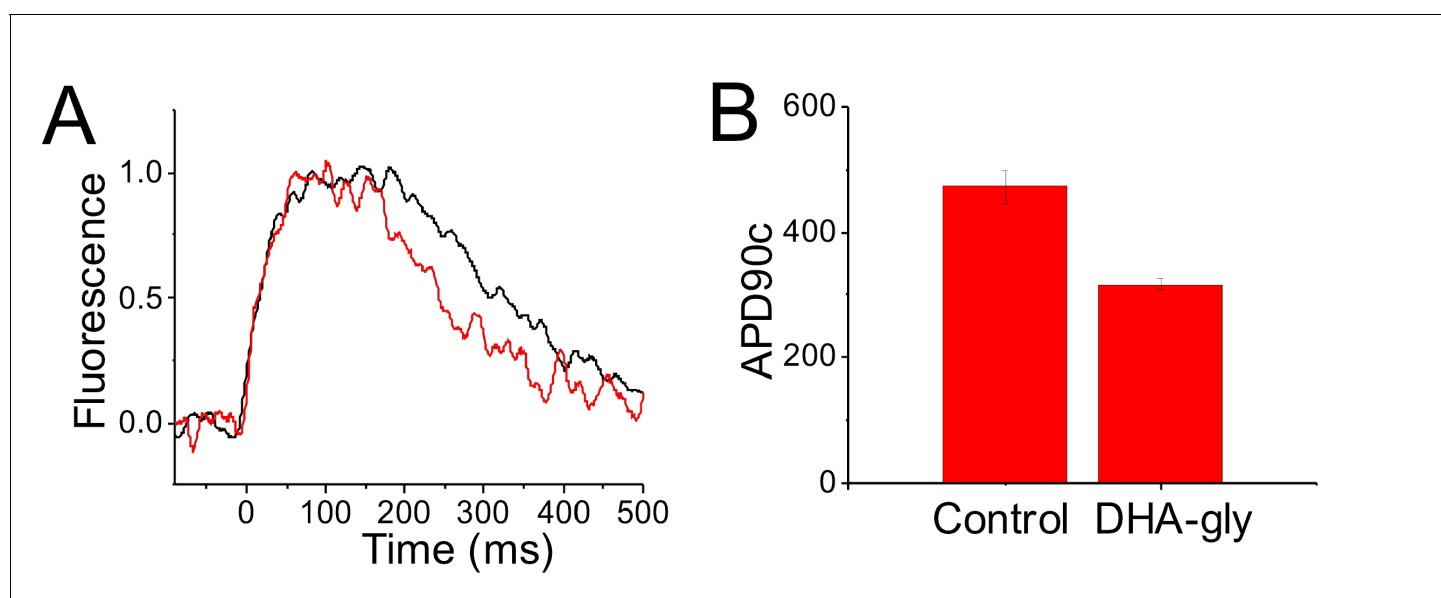

**Figure 8.** DHA-glycine decreases the APD90c of hiPSC cardiomyocytes. **(A)** Normalized representative CaT optical traces before (black) and after applied (red) 30 µM DHA-glycine on a monolayer of hiPSC-CM. **(B)** APD90c (ms) value in control conditions (black) and after applied (red) 30 µM DHA-glycine on hiPSC-CM (mean ± SEM; n = 3). *p<0.05.

analogues on individual channels, it opens up the possibility to target specific forms of LQTS in which specific channels are mutated.

In this work, we have demonstrated that PUFA analogues modulate several different voltage-gated ion channels, including some underlying the ventricular action potential: Kv7.1/KCNE1, Nav1.5/β1, and Cav1.2/β3/α2δ. The PUFA analogue-induced increases in current seen for Kv7.1/KCNE1 would tend to shorten the AP duration. We have previously shown that a PUFA analogue that increases $I_{Ks}$ currents shortens the AP duration and has anti-arrhythmic effects in rat cardiomyocytes (*Liin et al., 2016*). Work in animals has shown that PUFAs such as DHA and EPA are effective in terminating arrhythmia (*Kang and Leaf, 2000*). These anti-arrhythmic effects were proposed to occur through PUFA-mediated inhibition of $Na_V$ and $Ca_V$ channels, leading us to believe that PUFA analogue-induced inhibition of Cav1.2/β3/α2δ, and late Nav1.5/β1 currents would also tend to shorten the AP duration and potentially be anti-arrhythmic (*Kang and Leaf, 2000*; *Kang and Leaf, 1996*; *Xiao et al., 1997*; *Xiao et al., 1995*). Therefore, all of these effects could potentially be beneficial for patients with Long QT Syndrome. In the case of $I_{Ks}$ (Kv7.1/KCNE1) currents, PUFA analogues would rescue loss-of-function mutants of Kv7.1/KCNE1 ($I_{Ks}$) channels in LQT1 (KCNQ1 mutations) or LQT5 (KCNE1 mutations). In the case of $Na_V$ and $Ca_V$ currents, PUFAs would inhibit gain-of-function mutants of Nav1.5/β1 and Cav1.2/β3/α2δ channels in LQT3 or LQT8, respectively. In LQT3, it is increases in the late $Na_V$ currents that cause the prolonged APD. It is difficult to measure late Nav1.5/β1 currents in whole oocyte experiments. However, the PUFA-induced left-shifts in the voltage dependence of inactivation and decreases in the maximal conductance observed in our measured Nav1.5/β1 currents are assumed to also result in decreased Nav1.5/β1 late currents. There are several other currents, such as the $I_{Kr}$, that contribute to the ventricular action potential that we have not assessed here for potential PUFA effects. However, neither N-AT nor DHA-glycine up to 10 μM had any significant effects on currents through Herg channels (that together with the β subunit KCNE2 form the $I_{Kr}$ channels) expressed in heterologous systems (B.H. Bentzen and H. Duff, unpublished data).

Whether all these effects on these different ion channels are all anti-arrhythmic would have to be tested further in future in vitro and in vivo studies. However, to assess the potential for therapeutic effects of PUFA analogues on the cardiac action potential, we used the O'Hara-Rudy Dynamic (ORd) model in MATLAB to simulate the ventricular action potential in the presence of different PUFA analogues. In our simulations, PUFA analogues that are non-selective (i.e. that activate Kv7.1/KCNE1 while inhibiting Cav1.2/β3/α2δ and Nav1.5/β1) prevent the generation of an action potential. However, when we simulate the effects of Pin-glycine and DHA-glycine, which are both more selective for Kv7.1/KCNE1, we see a shortening in the action potential duration and the suppression of early afterdepolarizations. This suggests that selectively boosting $I_{Ks}$ (Kv7.1/KCNE1) current would be important for shortening and terminating the ventricular action potential. To further probe the therapeutic potential of PUFA analogues, we applied DHA-glycine to hiPSC-derived cardiomyocytes and found that DHA-glycine shortens the duration of the ventricular action potential. The next step in evaluating the therapeutic potential of PUFA analogues would be applying PUFA analogues to hiPSC-derived cardiomyocytes from patients bearing LQTS-causing mutations.

In our experiments using PUFA analogues on Nav1.5/β1, we observed both a shift in the voltage dependence of inactivation and a dose-dependent decrease in $Na^+$ currents. Extensive work has been done to characterize how each of the different voltage-sensing domains in $Na_V$ channels contribute to voltage-dependent activation and inactivation, many implicating DIV S4 in fast inactivation (*Capes et al., 2013*; *Ahern et al., 2015*). Recent work by *Hsu et al. (2017)* has also shown using voltage clamp fluorometry, the importance of both DIII and DIV in $Na_V$ channel inactivation (*Hsu et al., 2017*). Our data suggest that PUFA analogues may interact with S4 segments involved in voltage-dependent inactivation, allowing PUFA analogues to left-shift the voltage dependence of inactivation. However, this does not completely explain the additional dose-dependent decrease in $Na^+$ currents we observe on top of the leftward shifted voltage dependence of inactivation. Recent work by *Nguyen et al. (2019)* has uncovered a mechanism of $Na_V$ channel inhibition through a new pathway, allowing a hydrophobic molecules to permeate a fenestration between domains III and IV (DIII and DIV) in the human cardiac Nav1.5/β1 channel (*Nguyen et al., 2019*). It is possible that the hydrophobic PUFA analogue also block Nav1.5/β1 channels through this fenestration between DIII and DIV, causing the voltage-independent decrease in sodium currents.

The molecular mechanism of action of PUFA analogues on Cav1.2/β3/α2δ is still unclear, though we have shown that it does not occur through a shift in the voltage dependence of inactivation. In each case of Cav1.2/β3/α2δ inhibition by PUFA analogues, we observe a dose-dependent decrease in the $Ba^{2+}$ currents that appears as a decrease in $G_{max}$. There is evidence that some $Ca_V$ channel antagonists, such as dihydropyridines (DHPs) inhibit $Ca_V$ channels through an allosteric mechanism (*Tang et al., 2016*). *Pepe et al. (1994)* found that DHA alters the effectiveness of dihydropyridines, suggesting a shared binding site, or nearby binding sites, for DHPs and PUFAs (*Pepe et al., 1994*). *Tang et al. (2016)* found that dihydropyridines bind in a hydrophobic pocket near the pore of the bacterial $Ca_V$Ab channel and cause an allosteric conformational change that leads to disruption of the selectivity filter and thus inhibition of $Ca^{2+}$ currents (*Tang et al., 2016*). In addition, they observed that in the absence of DHPs a phospholipid occupies the DHP binding site (*Tang et al., 2016*). This would suggest that it is possible that PUFA analogues inhibit Cav1.2/β3/α2δ by binding to, or near, the DHP binding site and causing an allosteric conformational change that leads to a collapse of the pore and thus explaining the inhibition of Cav1.2/β3/α2δ currents without any changes in the voltage dependence of inactivation.

For simplicity, in this work we co-expressed Kv7.1, Cav1.2, and Nav1.5 with only some of the major auxiliary subunits that have high expression in ventricular cardiac cells to reflect the behavior of ion channel complexes in ventricular cardiomyocytes. For example, Kv7.1 was co-expressed with KCNE1, although other KCNE subunits and other auxiliary subunits, such as Yotiao (*Marx et al., 2002*), are also found in cardiomyocytes. Auxiliary subunit co-expression is known to modulate the pharmacology of different ion channels. For example, the co-expression of the β1 subunit with Nav1.5 alters PUFA-induced effects by DHA and α-linolenic acid (*Xiao et al., 2000*). However, we measured inhibition of Nav1.5/β1 by PUFA analogues with a taurine head group, showing that co-expression of the β1 subunit with Nav1.5 does not abolish all type of PUFA-induced inhibition. In addition, the Cav1.2 channel can be co-expressed with different β-subunits, including β3 (which we express in our experiments) and β2 (*Buraei and Yang, 2013*; *Hullin et al., 2003*). We found, though, that there are no differences between Cav1.2/β3/α2δ and Cav1.2/β2/α2δ in their response to Lin-taurine and Lin-glycine (*Figure 2—figure supplement 1*).

Work from several groups has demonstrated a shared electrostatic mechanism of action on voltage-gated $K^+$ channels (*Liin et al., 2018b*) and voltage-gated $Na^+$ channels (*Ahuja et al., 2015*) by biaryl sulfonamides. *Liin et al. (2018b)* showed that biaryl sulfonamides promote the activation of the Shaker $K^+$ channel through an electrostatic effect on the voltage sensing domain. In addition, *Ahuja et al. (2015)* showed that aryl sulfonamides inhibit $Na_V$ channels through an electrostatic 'voltage sensor trapping' mechanism that is specific for the Nav1.7 isoform. The work by Ahuja et al. supports the ability to pharmacologically target different ion channels with a high degree of selectivity (*Ahuja et al., 2015*). This is in agreement with our findings using PUFA analogues that show that PUFA analogues are variable in their channel selectivity, allowing us to target particular ion channels involved in the ventricular action potential.

It is possible that exogenously applied PUFAs cause changes in membrane properties that indirectly modulate the activity of membrane proteins, including ion channels (*G Lee, 2006*). For example, the length of the fatty acid acyl chain that incorporates into the membrane can lead to changes in the thickness of the membrane, which could indirectly alter ion channel activity (*G Lee, 2006*). The PUFA DHA is able to modulate the activity of gramicidin channels via changes in membrane fluidity (*Bruno et al., 2007*). One basis for the membrane fluidity hypothesis is that some studies found that PUFAs alter the activity of different ion channels at similar concentrations suggesting a similar mechanism (*Bruno et al., 2007*). However, in this work, we found significant differences in apparent binding affinity of some PUFA analogues for different voltage-gated ion channels expressed in the heart and that some PUFA analogues can modulate some channels selectively while having little to no effect on other cardiac ion channels. These data suggest that it is likely a direct effect on voltage-gated ion channel activity rather than an indirect effect through changes in membrane fluidity, although we cannot rule out that some PUFA analogue effects seen here are due to changes in membrane properties.

Our experiments were mainly conducted using the *Xenopus laevis* oocyte expression system at room temperature. The membrane composition differs between *Xenopus* oocytes and mammalian cells, leaving the possibility that *Xenopus* oocytes lack some signaling molecules and other factors that are present in cardiomyocytes under physiological conditions. For this reason, we conducted

experiments in human-induced pluripotent stem cell-derived cardiomyocytes which are more similar to native cardiomyocytes and these experiments were performed at physiological temperatures (37°). Doing this, we found that DHA-glycine is able to shorten the ventricular action potential as expected from experiments in *Xenopus* oocytes and simulated PUFA-induced effects on cardiomyocytes. We used a slightly higher dose of DHA-glycine (30 µM vs. 7 µM) and got a larger shortening of the APD90 (35% vs 4%) for the experiments on the cardiomyocytes compared to the oocyte experiments and the computer simulations. We tested lower doses of DHA-glycine on the cardiomyocytes but did not get significant effects at low doses. The reason for this is not clear, but it is possible that the cardiomyocytes do not incorporate as much PUFA analogues as oocytes or the affinities of the PUFA analogues for the different ion channels are slightly different in oocytes versus cardiomyocytes. In addition, neither the computer model nor the hiPSC-derived cardiomyocytes are perfect representations of the adult ventricular cardiomyocoytes and they could therefore respond slightly differently to the same stimulus. The main limiting factor with hiPSC-CMs is the level of maturation achieved (*Ronaldson-Bouchard et al., 2018*; *Scuderi and Butcher, 2017*). For example, hiPSC-CMs have lower levels of Nav1.5 and Kir2.1 channels (*Ronaldson-Bouchard et al., 2018*; *Verkerk et al., 2017*), which could reduce the upstroke velocity of the action potential and generate cardiomyocytes membrane potentials more positive than those usually observed in adult cardiomyocytes, respectively. The ORd computer simulation model has a significantly higher action potential plateau than in adult ventricular cardiomyocytes, the accommodation of the action potential duration in response to sodium changes in the ORd model is not in agreement with experimental data, and sodium channel block has a positive ionotropic effect in the Ord model instead of negative ionotropic effect (*Tomek et al., 2019*). In addition, the RYR channels dependence on calcium, the L-type $Ca^{2+}$ channels, and the $I_{Kr}$ channels are not ideally modelled in ORd (*Tomek et al., 2019*). An updated version of the ORd model that improve on some of these features was published during the review process of this paper (*Tomek et al., 2019*).

The work presented here demonstrates that PUFA analogues exert diverse modulatory effects on different types of voltage-gated ion channels through non-identical mechanisms. Because PUFA analogues modulate Kv7.1/KCNE1 channels through electrostatic effects, we hypothesized they would have similar effects on Cav1.2/β3/α2δ and Nav1.5/β1 channels. However, our data suggests that PUFA analogues can exert various modulatory effects on the activity of different ion channels, and that the mechanism depends on the ion channel that is being modulated. In addition, we have shown that PUFA analogues exhibit a range of selectivity for different ion channels, which depends both on the PUFA head group and the combination of PUFA head and tail groups. Using simulations of the ventricular action potential, we have shown that selective Kv7.1/KCNE1 channel activators are the most effective at shortening a prolonged ventricular action potential and suppressing early afterdepolarizations induced by hERG block. Last, in hi-PSC derived cardiomyocytes, we found that applying a PUFA analogue that is more selective for the Kv7.1/KCNE1 channel (DHA-glycine) is able to shorten the duration of the ventricular action potential. These data suggest that boosting Kv7.1/KCNE1 currents by using selective Kv7.1/KCNE1 channel activators can aid in restoring a normal action potential duration and prevent early afterdepolarizations.

## Materials and methods

Molecular Biology cRNA encoding Kv7.1 (UniProt: P51787), KCNE1 (UniProt: P15382.1), Nav1.5 (UniProt: Q14524.1), Nav β1 (UniProt: Q07699.1), Nav β3 (UniProt: Q07699.1), Cav1.2 (GenBank: CAA84346), Cav β3 (Uniprot: P54286), Cav β2 (Uniprot: P54288), and Cav α2δ (Uniprot: P13806) were transcribed using the mMessage mMachine T7 kit (Ambion). 50 ng of cRNA was injected into defolliculated *Xenopus laevis* oocytes (Ecocyte, Austin, TX): For Kv7.1/KCNE1 channel expression, we injected a 3:1, weight:weight (Kv7.1:KCNE1) cRNA ratio. For Nav1.5 channel expression, we injected a 2:1, weight:weight (Nav1.5:β1) cRNA ratio. For Cav1.2 channel expression, we injected a 2:1:1, weight:weight (Cav1.2:β3:α2δ) cRNA ratio. The exact stoichiometry for channel macromolecular complexes, including α -subunits and auxiliary subunits is not clear for many ion channels. For example, there have been multiple reported stoichiometries for Kv7.1:KCNE1 determined in heterologous expression systems with some groups reporting flexible stoichiometry of 4 Kv7.1: 1–4 KCNE1 and others reporting a fixed stoichiometry of 4 Kv7.1: 2 KCNE1 (*Murray et al., 2016*; *Nakajo et al., 2010*). The ratios of cRNA of α- and auxiliary subunits that we injected in *Xenopus* laevis oocytes are

such that all α-subunits are assumed to be in complex with auxiliary subunits. This allows a more homogenous population of ion channels with auxiliary subunits. One caveat to expressing ion channels as macromolecular complexes is that there are many other auxiliary subunits that could associate with the channel α-subunit that are present in cardiomyocytes and have the potential to alter PUFA-induced effects, but that are not being co-expressed in our experiments. In this work, we are co-expressing some of the auxiliary subunits that have been primarily reported to associate with their respective α-subunits in cardiomyocytes. Following cRNA injection, cells were incubated for 72–96 hr in standard ND96 solution (96 mM NaCl, 2 mM KCl, 1 mM $MgCl_2$, 1.8 mM $CaCl_2$, 5 mM HEPES; pH = 7.5) containing 1 mM pyruvate at 16°C prior to electrophysiological recordings.

## Two-electrode voltage clamp (TEVC)

*Xenopus laevis* oocytes were recorded in the two-electrode voltage clamp (TEVC) configuration. Recording pipettes were filled with 3 M KCl. The recording chamber was filled with ND96 (96 mM NaCl, 2 mM KCl, 1 mM $MgCl_2$, 1.8 mM $CaCl_2$, 5 mM HEPES; pH 7.5). For Cav1.2/β3/α2δ channel recordings, *Xenopus* oocytes were injected with 50 nl of 100 mM EGTA and incubated at 10°C for 30 min prior to electrophysiological recordings in order to sequester cytosolic calcium. In addition, Cav1.2/β3/α2δ channel recordings were done in $Ca^{2+}$-free solutions, using $Ba^{2+}$ as the charge carrier, to prevent calcium-dependent inactivation of Cav1.2/β3/α2δ channels. Calcium-dependent inactivation was not studied in this work, due to the presence of large calcium-activated chloride currents present in *Xenopus laevis* oocytes. PUFAs were obtained from Cayman Chemical (Ann Arbor, MI.) or synthesized in house (Linköping, Sweden) through methods previously described (*Bohannon et al., 2020*) and kept at −20 °C as 100 mM stock solutions in ethanol. Serial dilutions of the different PUFAs were prepared from stocks to make 0.2 μM, 0.7 μM, 2 μM, 7 μM, and 20 μM concentrations in ND96 solutions (pH = 7.5). PUFAs were perfused into the recording chamber using the Rainin Dynamax Peristaltic Pump (Model RP-1) (Rainin Instrument Co., Oakland, CA. USA).

Electrophysiological recordings were obtained using Clampex 10.3 software (Axon, pClamp, Molecular Devices). To measure Kv7.1/KCNE1 currents we apply PUFAs as the membrane potential is stepped every 30 s from −80 mV to 0 mV for 5 s before stepping to −40 mV and back to −80 mV to ensure that the PUFA effects on the current at 0 mV reached steady state. A voltage-step protocol was used to measure the current vs. voltage (I-V) relationship before PUFA application and after the PUFA effects had reached steady state for each concentration of PUFA. Cells were held at −80 mV followed by a hyperpolarizing prepulse to −140 mV. The voltage was then stepped from −100 to 60 mV (in 20 mV steps) followed by a subsequent voltage step to −20 mV to measure tail currents before returning to the −80 mV holding potential. For Cav1.2/β3/α2δ channel recordings, PUFAs are applied as the membrane potential is stepped from −80 mV to −30 mV and then 10 mV before returning to the holding potential of −80 mV. This allows the PUFA effects to reach steady state before recording voltage-dependent activation and inactivation. To measure voltage-dependent activation of Cav1.2/β3/α2δ, cells are held again at −80 mV and then stepped from −70 mV to 40 mV (in 10 mV steps). Voltage-dependent inactivation was measured by holding cells at −80 mV, applying a 500 ms conditioning pre-pulse at voltages between −80 mV and 20 mV (in 10 mV steps) before stepping to a test pulse of 10 mV to measure the remaining current and returning to −80 mV holding potential. For Nav1.5/β1 5 channel recordings, PUFAs are applied as the membrane potential is stepped from −80 mV to −90 mV for 480 ms before stepping to 30 mV for 50 ms and returned to a holding potential of −80 mV. This allows the PUFA effects to reach steady state before recording voltage-dependent activation and inactivation. To measure voltage-dependent activation of Nav1.5/β1, cells are held at −80 mV and then stepped from −90 mV to 40 mV (in 10 mV steps) and then returning to −80 mV holding potential. Voltage-dependent inactivation was measured by holding cells at −80 mV, applying a 500 ms conditioning pre-pulse at voltages between −140 mV and −30 mV (in 10 mV steps) and measuring the remaining current at a test pulse of −30 mV before returning to -80 mV holding potential.

## Data analysis

Tail currents from Kv7.1/KCNE1 measures were analyzed using Clampfit 10.3 software in order to obtain conductance vs. voltage (G-V) curves. The $V_{0.5}$, the voltage at which half the maximal current

occurs, was obtained by fitting the G-V curves from each concentration of PUFA with a Boltzmann equation:

$$G(V) = \frac{Gmax - Gmin}{1 + e^{\frac{V_{\frac{1}{2}} - V}{s}}} + Gmin$$

where $G_{max}$ is the maximal conductance at positive voltages and s is the slope factor in mV (**Table 2**). The current values for each concentration at 0 mV ($I/I_0$) were used to plot the dose response curves for each PUFA. These dose response curves were fit using the Hill equation in order to obtain the $K_m$ value for each PUFA:

$$\frac{I}{I_0} = 1 + \frac{A}{1 + \frac{Km^n}{x^n}}$$

where A is the fold increase in the current caused by the PUFA at saturating concentrations, $K_m$ is the apparent affinity of the PUFA, and n is the Hill coefficient. The maximum conductance ($G_{max}$) was calculated by taking the difference between the maximum and minimum current values (using the G-V curve for each concentration) and then normalizing to control solution (0 μM). In Cav1.2/β3/α2δ 2 and Nav1.5/β1 channels, peak currents (normalized to the peak values in control ND96) were used to determine PUFA induced changes in $I/I_0$, $\Delta V_{0.5}$ of inactivation, and $G_{max}$. Graphs plotting $I/I_0$, $\Delta V_{0.5}$, $G_{max}$, and $K_m$ were generated using the Origin 9 software (Northampton, MA.). To determine if there were significant differences between apparent binding affinity of individual PUFA analogues for Kv7.1/KCNE1, Cav1.2/β3/α2δ, or Nav1.5/β1 we conducted One-way ANOVA followed by Tukey's HSD for multiple comparisons when comparing all three channels or Student's t-test when comparing the apparent affinity for two channels. To determine if the PUFA-induced effects on $I/I_0$, $\Delta V_{0.5}$, or $G_{max}$ were statistically significant we conducted Student's t-test on the PUFA-induced effects at 7 μM. Significance α-level was set at $p < 0.05$ – asterisks denote significance: p<0.05*, p<0.01**, p<0.001***, p<0.0001****.

## Simulations

The effects of individual PUFA analogues were simulated on each ion channel using Berkeley Madonna modeling software and equations from the MATLAB code in the O'Hara and Rudy Dynamic (ORd) model (**O'Hara et al., 2011**). We individually simulated the Kv7.1/KCNE1, Cav1.2/β3/α2δ, and Nav1.5/β1 channels in Madonna and altered the parameters suggested to be modulated by PUFA binding to recapitulate our voltage clamp data from *Xenopus* oocytes. For example, to model the effects observed on the cardiac $I_{Ks}$ channel, we modified the voltage dependence of channel activation by shifting the $V_{0.5}$ as well as multiplying the $I_{Ks}$ conductance by the factor increase we observed in our experiments at a given PUFA concentration.

MATLAB simulations of the ventricular action potential in the epicardium of the heart were performed using the ORd model (**O'Hara et al., 2011**). To simulate the effects of PUFA analogues, we introduced the same modified parameters in the MATLAB code as we used to model the PUFA effects on the ionic currents in Berkeley Madonna. We made simultaneous changes to Kv7.1/KCNE1, Cav1.2/β3/α2δ, and Nav1.5/β1 for a given PUFA analogue and specific PUFA analogue concentration to model the effects of different PUFA analogues on the ventricular action potential under wild

**Table 2.** Fitted slope factor of conductance vs voltage relationship of Kv7.1/KCNE1 + PUFA analogues.

| PUFA Analogue | Control | 0.2 mM | 0.7 mM | 2 mM | 7 mM | 20 mM |
|---|---|---|---|---|---|---|
| Lin-taurine | 17.5 ± 0.75 mV | 17.7 ± 0.76 mV | 17.7 ± 0.77 mV | 18.6 ± 1.0 mV | 18.1 ± 1.0 mV | 12.2 ± 1.8 mV |
| N-AT | 16.7 ± 0.71 mV | 16.9 ± 0.65 mV | 17.2 ± 0.62 mV | 17.4 ± 0.60 mV | 18.2 ± 0.59 mV | 18.7 ± 0.73 mV |
| Pin-taurine | 17.7 ± 0.67 mV | 17.9 ± 0.82 mV | 17.7 ± 0.82 mV | 18.3 ± 0.96 mV | 18.8 ± 1.0 mV | 16.8 ± 0.73 mV |
| DHA-taurine | 18.7 ± 1.2 mV | 18.7 ± 1.1 mV | 18.6 ± 1.2 mV | 18.6 ± 1.3 mV | 18.1 ± 1.2 mV | NA |
| Lin-glycine | 18.4 ± 1.2 mV | 18.2 ± 0.89 mV | 19.0 ± 1.0 mV | 19.2 ± 1.2 mV | 20.6 ± 1.5 mV | 22.1 ± 1.7 mV |
| Pin-glycine | 18.4 ± 0.70 mV | 17.9 ± 0.83 mV | 17.8 ± 0.84 mV | 17.9 ± 0.91 mV | 18.7 ± 1.1 mV | 20.9 ± 1.4 mV |
| DHA-glycine | 16.7 ± 0.44 mV | 16.5 ± 0.49 mV | 16.6 ± 0.45 mV | 16.6 ± 0.52 mV | 17.0 ± 0.56 mV | 17.9 ± 0.68 mV |

type and LQTS conditions. To simulate susceptibility to early afterdepolarizations, hERG block by 0.1 µM dofetilide was simulated which previously has been shown to cause spontaneous early afterdepolarizations (*O'Hara et al., 2011*). To simulate the ability of PUFA analogues to suppress early afterdepolarizations, we altered the activity of Kv7.1/KCNE1, Cav1.2/β3/α2δ, and Nav1.5/β1 channels according to the PUFA-induced effects observed during experiments.

### Optical recordings of calcium transients in hiPSC-CM

Human induced pluripotent stem cell-derived cardiomyocytes (hiPSC-CM) were purchased from AXOL Bioscience. The hiPSC-CM were thawed and plated following the manufacturer's protocol. Briefly, cells were plated on Matrigel (Corning) pre-coated glass coverslips cultivated in 96-multiwell plates (Falcon). Matrigel was diluted in sterile DMEM/F12 medium (Gibco) (1:130) and incubated for 2 hr at 37°C before plating the cells. hiPSC-CM were cultured at 37°C and 5% $CO_2$. Cultured hiPSC-CM were fed every other day. To assess the calcium transients (CaT) with optical recordings, hiPSC-CM monolayers were loaded with the intracellular $Ca^{2+}$ indicator, Fluo-4FF AM (2 µM, Life-technologies) diluted in extracellular saline solution containing (in mM): 150 NaCl, 5.4 KCl, 1.8 $CaCl_2$, 1 $MgCl_2$, 15 Glucose, 15 HEPES, 1 Sodium pyruvate (pH 7.4; HCl) (*Ma et al., 2011*). This solution was also used for CaT optical recordings. All optical recordings were performed at 35–37°C. Stocks of DHA-Glycine (100 mM) were prepared in ethanol.

CaT fluorescence recordings from cell monolayers were recorded using a Leica DM LFS upright microscope. Light was focused on the sample through a 20X objective (20X/N.A. 0.8 Nikon S Fluor) and passed by a FITC filter cube (Chroma, HQ480/40; dichroic, Q505LP; and emitter, HQ535/50.). Fluorescence signals were passed through the emission filter, collected by a photodiode, and amplified with an Axopatch 200B patch clamp amplifier (Axon Instruments). Calcium fluorescence signals were low-pass Bessell-filtered (Frequency Devices) at 100 Hz, digitized at 1 kHz, and recorded using Clampex 9.2 and analyzed using Clampfit 9.2. CaT were used as an indirect measurement of the action potentials produced by hiPSC-CM. Since hiPSC-CM were spontaneously beating and at slightly different frequencies between monolayers, the frequency of the action potentials was calculated as the inverse of the time between CaTs. The action potential duration (APD) was measured in the absence and the presence of DHA-Glycine. The APD90 (or the time required to achieve the 90% of repolarization of the CaT) was measured in all the experiments to test the effects of DHA-glycine. APD90 was corrected in all measurements by a modified version of Fridericia's formula (*Fridericia, 2003*):

$$APDc = \frac{APD}{\left(\sqrt[3]{RR}\right)}$$

in order to reduce the APD90 variability induced by the different beating frequency due to the spontaneous beating of hiPSC-CM.

Ten CaT APD90c were measured and averaged in all the experimental conditions for each monolayer. The averaged CaT APD90 for each monolayer were corrected using Fridericia's formula and then averaged between monolayers. Data are presented as mean ± SEM. Statistical analysis as well as graphs were carried out with Origin9. Paired t-tests were performed and $p < 0.05$ was considered statistically significant.

## Acknowledgements

We thank Levi Lindroos, Victor Kornfeld, Siri Lundholm, and Sankhero Gewarges for their contributions during their time as visiting scholars.

## Additional information

### Competing interests

Sara I Liin, H Peter Larsson: A patent application (62/032,739) has been submitted by the University of Miami with SIL and HPL as inventors. The authors declare no other competing interests. The other authors declare that no competing interests exist.

## Funding

| Funder | Grant reference number | Author |
| --- | --- | --- |
| National Institutes of Health | R01-HL131461 | H Peter Larsson<br>Derek M Dykxhoorn |
| Swedish Research Council | 2017-02040 | Sara I Liin |

The funders had no role in study design, data collection and interpretation, or the decision to submit the work for publication.

## Author contributions

Briana M Bohannon, Alicia de la Cruz, Formal analysis, Investigation, Writing - original draft, Editing/ revision of manuscript; Xiaoan Wu, Formal analysis, Investigation; Jessica J Jowais, Investigation; Marta E Perez, Resources, Investigation; Derek M Dykxhoorn, Founding acquisition, Resources, Methodology; Sara I Liin, Conceptualization, Resources, Formal analysis, Funding acquisition, Investigation, Editing/revision of manuscript; H Peter Larsson, Conceptualization, Formal analysis, Supervision, Funding acquisition, Project administration, Editing/revision of manuscript

## Author ORCIDs

Briana M Bohannon (ID) https://orcid.org/0000-0002-3720-1477
Sara I Liin (ID) http://orcid.org/0000-0001-8493-0114
H Peter Larsson (ID) https://orcid.org/0000-0002-1688-2525

## Decision letter and Author response

Decision letter https://doi.org/10.7554/eLife.51453.sa1
Author response https://doi.org/10.7554/eLife.51453.sa2

# Additional files

## Supplementary files

- Transparent reporting form

## Data availability

Source data used in this manuscript is openly available and is listed as a source data file for accessibility.

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
