## [Decision Letter]

**Acceptance summary:**

By systematically studying the effects of various classes of PUFAs on different cardiac channels, your study identifies a specific class of PUFA that has the potential to act as an anti-arrhythmic.

**Decision letter after peer review:**

Thank you for submitting your work entitled "Polyunsaturated fatty acid analogues differentially affect cardiac Na_V_, Ca_V_, and K_V_ channels through unique mechanisms" for consideration by *eLife*. Your article has been reviewed by a Senior Editor, a Reviewing Editor, and three reviewers. The following individual involved in review of your submission has agreed to reveal their identity: Jon T Sack (Reviewer #3).

This manuscript describes the mechanisms of differential modulation of cardiac ion channels by PUFAs. By testing various PUFA analogs, your study finds that while many PUFAs broadly modify the cardiac voltage-gated ion channels, those with glycine headgroup are more specific for Kv7.1/KCNE1 channels. Simulations of cardiac models suggest that these compounds can act as potential anti-arrhythmics. This is the most exciting aspect of this study but the lack of experimental demonstration of this application in cardiomyocytes (or equivalent model systems) significantly dampened the enthusiasm of the reviewers.

*Reviewer #1:*

The manuscript entitled "Polyunsaturated fatty acid analogues differentially affect cardiac Na_V_, Ca_V_, and K_V_ channels through unique mechanisms" by Bohannon et al. describes the effect of PUFA analogues on the gating properties of Kv7.1/KCNE1, Cav1.2/beta3/alpha2gamm, and Nav1.5/beta1 channel complex. The authors find that synthesized PUFA analogues have different gating effects and apparent affinities for each of the channel complexes. Based on their experimental data and cardiomyocyte simulations, the authors proposed that the studied PUFA analogs can be used to treat specific sub-types of Long-QT syndrome.

Although the data and analyses are very robust, the interpretations/conclusions in some cases did not match the results. The introduction and discussion can be improved to make the story more comprehensive. Here are some comments, that hopefully will help improving the manuscript:

1) Use the Introduction to explain why the effects of these PUFA analogs are being tested on the channel complexes at the particular ratios used. I am guessing, the pore domain-auxiliary subunit ratios have been determined elsewhere, and this has to be clearly addressed to guide the reader. Furthermore, authors should address if these protein ratios are representative of what it is found in cardiomyocytes.

2) Comment 1 also prompts the question if the PUFA analogs have differential effects on the pore domain and/or accessory subunits. For instance, Hoshi et al., published two manuscripts in which they reported that BK channel function is modified by DHA and this effect (Hoshi et al., 2013) is enhanced by co-expressing along with the beta1 or beta4 subunits. Is this the case for any of these complexes? If this has been published it should be mentioned in the Introduction or as an open question in the Discussion section.

3) In the Introduction the authors say "Though both PUFAs and PUFA analogues are known to modulate different ion channel activities (i.e. processes underlying activation inactivation), it is unclear whether specific PUFAs and PUFA analogues are selective for certain ion channels or if they broadly influence the activity of several different ion channels simultaneously". This prompts to wonder if when in cardiomyocytes or when co-expressed in oocytes, these effects would be similar to the ones reported when only overexpressing one channel complex.

4) The statement in subsection “PUFA analogues modulate Kv7.1/KCNE1, Cav1.2, and Nav1.5 through distinct mechanisms” reads: "We have found through previous work that PUFA analogues with taurine head groups are good activators of the Kv7.1/KCNE1 channels due to the low pKa of the taurine head group (37, 38). Having a lower pKa allows the taurine head group to be fully negatively charged at physiological pH so that it has maximal electrostatic effects on Kv7.1/KCNE1 channels (38)". This published conclusion is completely overruled by the results shown on Figure 5C. Comparing results on Figure 5C and Figure 4B strongly suggests that is the acyl chain and not the taurine group what modifies Kv7.1/KCNE1 voltage dependence. Pin-taurine and DHA-taurine also add to the idea that is the acyl chain and not the taurine group what affects Kv7.1/KCNE1 complex gating. Can the authors elaborate on this?

5) Without some experimental data in cardiomyocytes, the simulation lacks validity, as the authors rightfully pointed out that oocytes are not cardiomyocytes (e.g., different lipid and protein composition, as well as their plasma membrane distribution). Moreover, overexpression of both channel and auxiliary subunits further complicate this. These channel complexes are expressed in the same cardiomyocytes: hence, perfusing fatty acids to cardiomyocytes with different binding affinities for the used complexes might elicit a more complex responses than anticipated.

*Reviewer #2:*

Bohannon et al., present a well-written manuscript describing the pharmacological effects of polyunsaturated fatty acid (PUFA) analogues on three major cardiac voltage-gated ion channels (NaV, CaV, KV). PUFA has been proposed as a potential anti-arrhythmic strategy. However, PUFA therapy requires understanding how PUFA analogues differentially affect distinct cardiac channels. To this end, the authors measured dose-dependent effect of PUFA with different head and tail groups on NaV1.5, CaV1.2, and KV7.1/KCNE1. The functional readouts include voltage-dependence of activation/inactivation and maximum conductance. The results reveal that: (1) different PUFA analogues affect the major cardiac channels through different mechanisms and with different IC50s, (2) the head group (glycine vs. taurine) specifies broad vs. specific channel modulatory effects, and (3) PUFAs selective for KV7.1/KCNE1 but not NaV1.5 or CaV1.2 might be anti-arrhythmic.

This research group has previously published a body of work dissecting the mechanism of PUFA regulation of KV7.1/KCNE1. This manuscript provides further insights for how these mechanisms relate to NaV1.5 and CaV1.2 and is significant. This manuscript establishes an important proof-of-concept for a PUFA (DHA-glycine) selective for one cardiac channel (KV7.1/KCNE1) and demonstrates its anti-arrhythmic potential in silico. This is a critical step toward clinically applying PUFA. The manuscript is overall very clear, nicely presented, and well-written. However, the reliance on heterologous expression systems to test the effects without follow on in a native or native-like myocyte somewhat limits confidence in the predicted therapeutic benefits.

Essential revisions:

1) The authors expressed β3 subunit with CaV1.2 channels in this study. However, the dominant CaVβ subunit in the heart is β2. β3 and β2 subunits are known to differentially affect CaV1.2 voltage-dependent inactivation. It is unclear how PUFA may affect CaV1.2 inactivation when associated with β3 vs. β2. Given this manuscript's focus on cardiac channels, how β2 vs. β3 could change PUFA's effect on CaV1.2 inactivation should be addressed.

2) As exemplified in the previous point, recapitulating the native macromolecular complexes of these channels in heterologous expression systems can be challenging and may significantly affect the outcomes of the study. These challenges may be addressed by testing compounds in native or iPSC-derived cardiomyocytes where the effects on APD could be tested.

3) Subsection “PUFA analogues modulate Kv7.1/KCNE1, Cav1.2, and Nav1.5 through distinct mechanisms”. Figure 6E-G: the authors claim that pin-taurine inhibits CaV1.2 current, but pin-taurine does not significantly affect voltage-dependence of inactivation or decrease Gmax. Can the authors explain how CaV1.2 current is inhibited by pin-taurine at 7 μM?

4) The authors indicate that PUFA inhibitory effect on NaV1.5 can be potential therapy for LQT3. LQT3 typically arises from increase in NaV1.5 late current which prolongs APD. The effect of different PUFA on NaV1.5 late current is not reported. The authors should revisit the NaV1.5 data for late current analysis. It would be illuminating to determine if PUFA exert any regulation on NaV1.5 late current.

5) In general, the strong focus on LQT syndrome, which is quite rare, is exclusive of more prevalent arrhythmias resulting from drug-induced LQT syndrome, ischemia and heart failure. A potential connection to these pathologies would increase the potential for PUFA-derived therapies to translate.

6) IKs current should be strongly rate-dependent in the ORd computational model. Given that DHA-glycine shows specificity for IKs, the authors should test the effect at different cycle lengths to examine its effect on APD rate dependence.

*Reviewer #3:*

This manuscript addresses an interesting question of ion channel modulator polypharmacology in the context of voltage-gated cardiac ion channels. The major finding of this work is that some PUFAs which inhibit Nav1.5 and Cav1.2 channels also activate Kv7.1/KCNE channels. The results indicate that a series of PUFA variants have differing modulatory effects on selected voltage gated ion channels that undergird action potentials in human ventricular myocytes. The effect of PUFA modulation of these 3 ion channels were input into cardiac action potential simulations, resulting in distinct effects on the cardiac action potential waveform.

The experimental methods appear to be appropriately transparently reported. The electrophysiology appears to be of high quality, using *Xenopus* oocyte 2-electrode clamp.

Essential revisions:

1) It is intriguing that the range of concentrations where the Nav1.5 and Kv7 activities occurs are often similar. Possibly a certain critical membrane mole fraction of partitioned PUFA is needed for both effects? It is also intriguing that the magnitude of effects of different PUFAs on Nav1.5 and Cav1.2 seem to be correlated for different PUFA. The non-selective effects of PUFAs on sodium channels appear consistent with perturbation of bulk membrane bilayer properties. PUFAs modulate membrane protein function by bilayer-mediated mechanisms that do not involve specific protein binding but rather changes in bilayer material properties. Other compounds that similarly alter bilayer material properties result in inhibition and a negative voltage shift in the steady state inactivation of voltage-gated Na channels. This possibility could be investigated, along with the implication of this mechanism: the modulation of membrane protein activity of the PUFAs may not be limited to Nav1.5 and Cav1.2 but also extend to other Kv, mechanosensitive and many other classes of membrane proteins that are modulated by the physical properties of lipid bilayers. See PMID: 17535898, 15111647, 15967874, 24901212.

2) In the cardiac action potential simulations, the PUFAs modulate only Nav1.5, Cav1.2 and Kv7.1/KCNE channels. However, given that the PUFAs are non-selective modulators it seems reasonable that they would modulate additional currents that undergird the cardiac action potential. Given this, the simulations do not appear to form a strong basis for the inferences about the anti-arrhythmogenic effects of the PUFAs. Speculation about arrhythmogenic effects could be better distinguished from results and inescapable interpretations.

3) The analysis methods require further clarification.

[Editors’ note: further revisions were suggested prior to acceptance, as described below.]

Thank you for resubmitting your work entitled "Polyunsaturated fatty acid analogues differentially affect cardiac Na_V_, Ca_V_, and K_V_ channels through unique mechanisms" for further consideration by *eLife*. Your revised article has been evaluated by Olga Boudker (Senior Editor) and a Reviewing Editor.

The manuscript has been improved but there are some remaining issues that need to be addressed before acceptance, as outlined below:

1) "The increased percent shortening at higher bpm is most likely because the interval between action potentials is not long enough at high bpm to move all Kv7.1/KCNE1 channels into deep closed states, which leads to an rate-dependent accumulation of open Kv7.1/KCNE1 channels (46). The increased percent shortening at lower bpm is most likely due to that Kv7.1/KCNE1 channels have more time to open during the longer action potentials at low bpm." – These are reasonable comments. However, with a model it should be straightforward to make a definitive conclusion rather than pointing to a likely mechanism.

2) Do the PUFAs affect hERG? Mentioning this or stating that it is unknown, yet important, seems critical.

3) Limitations of testing predictions of the O'Hara-Rudy ventricular myocyte model in iPSC derived myocytes should be discussed.

4) In this manuscript, therapeutic implications are extrapolated from a limited basis. e.g in the Abstract: "Our data suggest that PUFA analogues could be effective therapeutics for LQTS and cardiac arrhythmia." Wording to convey what the PUFAs do in a reduced system while refraining from speculating in the Results section about their therapeutic potential effects in a more complex system would seem prudent. Caution seems especially warranted given non-specificity or dirtiness of a PUFA approach; this could be given more discussion. Striking or carefully qualifying statements in the abstract about the therapeutic potential of PUFAs could help readers avoid the impression that therapeutic potential is tested in this manuscript.

5) Abstract:

"In addition, PUFA analogues that are selective for the cardiac IKs channel (Kv7.1/KCNE1) are effective in shortening the cardiac action potential in human-induced pluripotent stem cell-derived cardiomyocytes."

Only one PUFA analogue was tested on cardiomyocytes.

6) Introduction:

Description of cardiac APs is oversimplified. Species is not explicitly defined. Many current types are ignored.

7) Subsection “PUFA analogues with taurine head groups are non-selective and broadly modulate multiple cardiac ion channels, with preference for Nav1.5/β1” "selectivity of N-AT for cardiac ion channels and at more therapeutically feasible concentrations." What are therapeutically feasible concentrations?

8) Discussion section: "The effects of PUFA analogues on Kv7.1/KCNE1, Cav1.2/β3/α2δ, and Nav1.5/β1 individually are anticipated to have antiarrhythmic effects and would potentially be beneficial for patients with Long QT Syndrome". Basis for this statement?

9) Note: labeling figures each with its figure # could make the job of the reviewers easier.

10) Figure 1: "PUFA analogue, Linoleoyl-taurine, activates Kv7.1/KCNE1 channels through an electrostatic mechanism on voltage sensor and pore."

The figure itself does not show that an electrostatic mechanism on voltage sensor and pore is involved.

11) Figure 2—figure supplement 1: panels C and D are labeled Lin-glycine, while legend states Lin-taurine

12) Table 1: the Km and A values from many of the fits in Figure 4, Figure 5 and Figure 6 don't seem to be meaningful, as they are underconstrained when dose-responses don't saturate. A statistic for Table I that reflects the error of the fits themselves (e.g. standard deviation representing the confidence of the least mean squared fitting) could be more meaningful.

13) Figure 4 and Figure 5: It is still a bit murky as to how, precisely Gmax/Gmax0 dose-response curves were fit. At low concentrations, the curves don't always plateau at Gmax/Gmax0=1.

---

## [Author Response]

[…] but the lack of experimental demonstration of this application in cardiomyocytes (or equivalent model systems) significantly dampened the enthusiasm of the reviewers.We here provide new data that shows that DHA-glycine shortens the APD in human iPSC-derived cardiomyocytes (New Fig. zz), consistent with our APD simulations. We have also previously shown that DHA-glycine shortens the APD in excised guinea pig hearts (Lin et al., 2015).Reviewer #1:[…] Although the data and analyses are very robust, the interpretations/conclusions in some cases did not match the results. The Introduction and Discussion section can be improved to make the story more comprehensive. Here are some comments, that hopefully will help improving the manuscript:1) Use the Introduction to explain why the effects of these PUFA analogs are being tested on the channel complexes at the particular ratios used. I am guessing, the pore domain-auxiliary subunit ratios have been determined elsewhere, and this has to be clearly addressed to guide the reader. Furthermore, authors should address if these protein ratios are representative of what it is found in cardiomyocytes.

We now explain the rationale behind the co-injected RNAs in the Materials and methods section. For many channel complexes the exact stoichiometry is not known, especially in native tissues. However, even for channel complexes for which stoichiometries have been measured, for example Kv7.1/KCNE1, there are conflicting results. For example, one study showed a fixed 4:2 stoichiometry, whereas another study showed a flexible stoichiometry. Neither of these measurements was done in native tissues, so it still unclear what the stoichiometry is in ventricular cells. We overexpress the β subunits to obtain a homogeneous channel population, under the assumption that for the channel complexes in cardiomyocytes all α subunits are bound to accessory subunits. This assumption is now clearly stated. We also mention the caveat that there are many more accessory subunits in cardiomyocytes that could associate with the channels and potentially alter the PUFA effects.

2) Comment 1 also prompts the question if the PUFA analogs have differential effects on the pore domain and/or accessory subunits. For instance, Hoshi et al., published two manuscripts in which they reported that BK channel function is modified by DHA and this effect (Hoshi et al., 2013) is enhanced by co-expressing along with the beta1 or beta4 subunits. Is this the case for any of these complexes? If this has been published it should be mentioned in the Introduction or as an open question in the Discussion section.

We mention in the Introduction that we have previously shown that the KCNE1 subunit abolishes the effect of DHA, but not those of DHA-glycine and N-AT, by promoting protonation of the carboxyl group of DHA (Liin et al., 2015). We now include a paragraph in the Discussion section about possible effects of accessory subunits.

3) In the Introduction the authors say "Though both PUFAs and PUFA analogues are known to modulate different ion channel activities (i.e. processes underlying activation inactivation), it is unclear whether specific PUFAs and PUFA analogues are selective for certain ion channels or if they broadly influence the activity of several different ion channels simultaneously". This prompts to wonder if when in cardiomyocytes or when co-expressed in oocytes, these effects would be similar to the ones reported when only overexpressing one channel complex.

We agree that the effects in a cardiomyocyte could be different than in an oocyte overexpressing the channel complexes. We bring this point up in the Discussion section. We now also show PUFA effects on human cardiomyocytes (New Figure 8) and we have previously shown PUFA effects on whole guinea pig hearts.

We don’t really understand the second point about effects on only one channel complex versus a combination of channel complexes (If we understand the reviewer’s point correctly?). We assume that the different channel complexes are not directly interacting with each other and therefore the individual effects on the channel parameters would just be additive in a cardiomyocyte (under the assumption that each channel complex responds the same way in an oocytes as in a cardiomyocyte). Of course, the effect on the action potential (AP) is more complex and all effects from all channel complexes have to be considered to get a realistic picture of what PUFAs will do to the AP. This is why we did the computer simulations in the first place, since it is virtually impossible to guess what the effect on the AP will be without taking the effects on all channels into account.

4) The statement in subsection “PUFA analogues modulate Kv7.1/KCNE1, Cav1.2, and Nav1.5 through distinct mechanisms” reads: "We have found through previous work that PUFA analogues with taurine head groups are good activators of the Kv7.1/KCNE1 channels due to the low pKa of the taurine head group (37, 38). Having a lower pKa allows the taurine head group to be fully negatively charged at physiological pH so that it has maximal electrostatic effects on Kv7.1/KCNE1 channels (38)". This published conclusion is completely overruled by the results shown on figure 5C. Comparing results on Figure 5C and Figure 4B strongly suggests that is the acyl chain and not the taurine group what modifies Kv7.1/KCNE1 voltage dependence. Pin-taurine and DHA-taurine also add to the idea that is the acyl chain and not the taurine group what affects Kv7.1/KCNE1 complex gating. Can the authors elaborate on this?

We understand how the reviewer got to this conclusion from old Figure 5C and Figure 4B (now New Figure 4A-B). However, the reviewer missed maybe that we have shown before that N-AT has a big robust effect on Kv7.1/KCNE1, but only at higher doses. Here we only used low concentrations. We are sorry for the confusion and now we state this more clearly. We have previously published several papers in which we clearly show that it is the charge on the head group that determines the effect on Kv7.1/KCNE1 (Liin et al., 2015; Larsson et al., 2018; Liin et al., 2018). However, the acyl tail affects the affinity of the PUFA for the Kv7.1/KCNE1 channel (Bohannon et al., 2018).

5) Without some experimental data in cardiomyocytes, the simulation lacks validity, as the authors rightfully pointed out that oocytes are not cardiomyocytes (e.g., different lipid and protein composition, as well as their plasma membrane distribution). Moreover, overexpression of both channel and auxiliary subunits further complicate this. These channel complexes are expressed in the same cardiomyocytes: hence, perfusing fatty acids to cardiomyocytes with different binding affinities for the used complexes might elicit a more complex responses than anticipated.

We now show effects in cardiomyocytes that are consistent with our simulations (Figure 8).

We don’t believe that the PUFAs are the limiting factor in the oocyte experiments or in cardiomyocytes (i.e. that there are so many channel complexes in the cell that the channels will act like a PUFA chelator). A concentration of μM of PUFAs have been estimated to give a surface density of PUFAs in cells in the >1000/μm^2^ (Pound, Kang and leaf, 2001). The estimated density of channels in a cell is much lower, around 1-10/μm^2^ (except at the node of Ranvier or electric plaques in electric eel), so the number of PUFAs will most likely not be a limiting factor in our experiments.

Reviewer #2:Bohannon et al., present a well-written manuscript describing the pharmacological effects of polyunsaturated fatty acid (PUFA) analogues on three major cardiac voltage-gated ion channels (NaV, CaV, KV). PUFA has been proposed as a potential anti-arrhythmic strategy. However, PUFA therapy requires understanding how PUFA analogues differentially affect distinct cardiac channels. To this end, the authors measured dose-dependent effect of PUFA with different head and tail groups on NaV1.5, CaV1.2, and KV7.1/KCNE1. The functional readouts include voltage-dependence of activation/inactivation and maximum conductance. The results reveal that: (1) different PUFA analogues affect the major cardiac channels through different mechanisms and with different IC50s, (2) the head group (glycine vs. taurine) specifies broad vs. specific channel modulatory effects, and (3) PUFAs selective for KV7.1/KCNE1 but not NaV1.5 or CaV1.2 might be anti-arrhythmic.This research group has previously published a body of work dissecting the mechanism of PUFA regulation of KV7.1/KCNE1. This manuscript provides further insights for how these mechanisms relate to NaV1.5 and CaV1.2 and is significant. This manuscript establishes an important proof-of-concept for a PUFA (DHA-glycine) selective for one cardiac channel (KV7.1/KCNE1) and demonstrates its anti-arrhythmic potential in silico. This is a critical step toward clinically applying PUFA. The manuscript is overall very clear, nicely presented, and well-written. However, the reliance on heterologous expression systems to test the effects without follow on in a native or native-like myocyte somewhat limits confidence in the predicted therapeutic benefits.

See comment to editor above and new Figure 8.

Essential revisions:1) The authors expressed β3 subunit with CaV1.2 channels in this study. However, the dominant CaVβ subunit in the heart is β2. β3 and β2 subunits are known to differentially affect CaV1.2 voltage-dependent inactivation. It is unclear how PUFA may affect CaV1.2 inactivation when associated with β3 vs. β2. Given this manuscript's focus on cardiac channels, how β2 vs. β3 could change PUFA's effect on CaV1.2 inactivation should be addressed.

We now have done some experiments with the beta2 subunit to confirm that the effects looks similar as with beta3 (shown in new Figure 2—figure supplement 1). Since PUFAs do not affect CaV1.2 inactivation, but just reduce the maximum conductance, one would not expect any differences using beta2 or beta3 (unless there were some direct steric interference of the β subunits, which doesn’t seem to be the case since we seem similar effects with beta2 and beta3).

2) As exemplified in the previous point, recapitulating the native macromolecular complexes of these channels in heterologous expression systems can be challenging and may significantly affect the outcomes of the study. These challenges may be addressed by testing compounds in native or iPSC-derived cardiomyocytes where the effects on APD could be tested.

We now show effect of PUFAs on cardiomyocytes in new Figure 8 (see comment to editor above).

3) Subsection “PUFA analogues modulate Kv7.1/KCNE1, Cav1.2, and Nav1.5 through distinct mechanisms””. Figure 6E-G: the authors claim that pin-taurine inhibits CaV1.2 current, but pin-taurine does not significantly affect voltage-dependence of inactivation or decrease Gmax. Can the authors explain how CaV1.2 current is inhibited by pin-taurine at 7 μM?

Sorry for the misleading text. Pin-taurine does significantly inhibit CaV1.2 at 20uM, not at 7uM. This is now better explained in the text.

4) The authors indicate that PUFA inhibitory effect on NaV1.5 can be potential therapy for LQT3. LQT3 typically arises from increase in NaV1.5 late current which prolongs APD. The effect of different PUFA on NaV1.5 late current is not reported. The authors should revisit the NaV1.5 data for late current analysis. It would be illuminating to determine if PUFA exert any regulation on NaV1.5 late current.

It is very hard to measure effects on late currents in TEVC recordings of NaV1.5. The NaV1.5 currents must preferably be smaller than 1 uA in TEVC recordings in *Xenopus oocytes* to have good voltage clamp. This means that the late currents would be in the order of 10 nA and changes in the late currents would be even smaller, which is too small to reliably record in TEVC. However, the two PUFA effects on NaV1.5 currents (a reduction in overall conductance and a negative shift in the inactivation curve) are both expected to reduce the late currents. We now state explicitly this assumption.

5) In general, the strong focus on LQT syndrome, which is quite rare, is exclusive of more prevalent arrhythmias resulting from drug-induced LQT syndrome, ischemia and heart failure. A potential connection to these pathologies would increase the potential for PUFA-derived therapies to translate.

It is not obvious to us how to make these connections, except for drug-induced LQT syndrome. And even then, most people don’t like when we say that PUFAs can be used to treat drug-induced LQTS, because FDA will not approve combination therapies in most cases. It would be even harder to make a case for ischemia and heart failure. There is an APD prolongation in heart failure, so PUFAs could potentially help here. But ischemia and heart failure are a lot more complex that Long QT Syndrome. So, at this point, we are not comfortable to propose anything else than to use PUFAs to treat Long QT Syndrome, which is more directly related to our data.

6) IKs current should be strongly rate-dependent in the ORd computational model. Given that DHA-glycine shows specificity for IKs, the authors should test the effect at different cycle lengths to examine its effect on APD rate dependence.

We now show the effect of DHA-glycine at different rates in Figure 7—figure supplement 1.

Reviewer #3:[…] The experimental methods appear to be appropriately transparently reported. The electrophysiology appears to be of high quality, using *Xenopus* oocyte 2-electrode clamp.Essential revisions:1) It is intriguing that the range of concentrations where the Nav1.5 and Kv7 activities occurs are often similar. Possibly a certain critical membrane mole fraction of partitioned PUFA is needed for both effects? It is also intriguing that the magnitude of effects of different PUFAs on Nav1.5 and Cav1.2 seem to be correlated for different PUFA. The non-selective effects of PUFAs on sodium channels appear consistent with perturbation of bulk membrane bilayer properties. PUFAs modulate membrane protein function by bilayer-mediated mechanisms that do not involve specific protein binding but rather changes in bilayer material properties. Other compounds that similarly alter bilayer material properties result in inhibition and a negative voltage shift in the steady state inactivation of voltage-gated Na channels. This possibility could be investigated, along with the implication of this mechanism: the modulation of membrane protein activity of the PUFAs may not be limited to Nav1.5 and Cav1.2 but also extend to other Kv, mechanosensitive and many other classes of membrane proteins that are modulated by the physical properties of lipid bilayers. See PMID: 17535898, 15111647, 15967874, 24901212.

We are well aware of these publications and potential membrane effects. However, we have previously shown that negative and positive PUFAs have opposite effects on Kv7.1/KCNE1 (and neutral PUFAs have no effect) and altering external pH to protonate and neutralize the head group removes the effects of PUFAs. In addition, removing specific charges on Kv7.1 removes the effects of PUFAs without significantly altering the behavior of Kv7.1 under control conditions. The apparent affinity of PUFA is also altered by specific mutations on Kv7.1. These findings would be hard to explain by indirect membrane effects, but very simple to explain by direct binding and an electrostatic effect on the Kv7.1/KCNE1 channel. In addition, we show in new Figure 6 that some PUFAs different concentrations are needed for the effects on the different channels, as if, at least, some effects are due to direct effects on the channels and not due to general membrane effects. However, we cannot exclude that some of the effects are due to membrane effects. We now include a paragraph about additional potential membrane effects on ion channels.

2) In the cardiac action potential simulations, the PUFAs modulate only Nav1.5, Cav1.2 and Kv7.1/KCNE channels. However, given that the PUFAs are non-selective modulators it seems reasonable that they would modulate additional currents that undergird the cardiac action potential. Given this, the simulations do not appear to form a strong basis for the inferences about the anti-arrhythmogenic effects of the PUFAs. Speculation about arrhythmogenic effects could be better distinguished from results and inescapable interpretations.

We now show effects of DHA-glycine on cardiomyocytes in Figure 8, which support the computer simulations.

[Editors’ note: further revisions were suggested prior to acceptance, as described below.]

The manuscript has been improved but there are some remaining issues that need to be addressed before acceptance, as outlined below:1) "The increased percent shortening at higher bpm is most likely because the interval between action potentials is not long enough at high bpm to move all Kv7.1/KCNE1 channels into deep closed states, which leads to an rate-dependent accumulation of open Kv7.1/KCNE1 channels (46). The increased percent shortening at lower bpm is most likely due to that Kv7.1/KCNE1 channels have more time to open during the longer action potentials at low bpm." – These are reasonable comments. However, with a model it should be straightforward to make a definitive conclusion rather than pointing to a likely mechanism.

We now show the IKs current at the different bpm and the rate-dependent accumulation of IKs channels in the 200 bpm simulation (Figure 8—figure supplement 2). However, we do not find any clear reason for the increased percent shortening at 40 bpm. We now have changed the text as follows: “At high bpm, the IKS current increases due to a rate-dependent accumulation of open Kv7.1/KCNE1 channels (46) (Figure 8—figure supplement 2), which most likely underlies the larger percent shortening of the APD90 at 200 bpm. The reason for the increased percent shortening at 40 bpm compared to 60 bpm is not clear (Figure 8—figure supplement 2).” Note that this is a very non-linear system (non-voltage clamp system with intracellular factors, such as intracellular calcium and calmodulin, able to change and alter channel parameters) and it is therefore not easy to understand what is cause and effect in this system.

2) Do the PUFAs affect hERG? Mentioning this or stating that it is unknown, yet important, seems critical.

We and our collaborators have tested NAT and DHA-glycine on hERG channels and have not seen any significant effects. This is now mentioned in the Discussion section as unpublished data. We are currently working on testing the other PUFA compounds on hERG for another manuscript.

3) Limitations of testing predictions of the O'Hara-Rudy ventricular myocyte model in iPSC derived myocytes should be discussed.

We now list the limitations of the computer models as well as the IPSC-derived cardiomyocytes in the Discussion section.

4) In this manuscript, therapeutic implications are extrapolated from a limited basis. e.g in the Abstract: "Our data suggest that PUFA analogues could be effective therapeutics for LQTS and cardiac arrhythmia." Wording to convey what the PUFAs do in a reduced system while refraining from speculating in the Results section about their therapeutic potential effects in a more complex system would seem prudent. Caution seems especially warranted given non-specificity or dirtiness of a PUFA approach; this could be given more discussion. Striking or carefully qualifying statements in the abstract about the therapeutic potential of PUFAs could help readers avoid the impression that therapeutic potential is tested in this manuscript.

We have reworded the Abstract and Results section as suggested.

5) Abstract:"In addition, PUFA analogues that are selective for the cardiac IKs channel (Kv7.1/KCNE1) are effective in shortening the cardiac action potential in human-induced pluripotent stem cell-derived cardiomyocytes."Only one PUFA analogue was tested on cardiomyocytes.

Changed as suggested.

6) Introduction:Description of cardiac APs is oversimplified. Species is not explicitly defined. Many current types are ignored.

A more thorough description of human cardiac APs is now described with some more current types added.

7) Subsection “PUFA analogues with taurine head groups are non-selective and broadly modulate multiple cardiac ion channels, with preference for Nav1.5/β1” "selectivity of N-AT for cardiac ion channels and at more therapeutically feasible concentrations." What are therapeutically feasible concentrations?

NAT only shows effect for IKs channels for concentrations >30 μM, which is close to the critical micelle concentration (70-200 μM). Something below 10 μM would be more reasonable as a therapeutic. We have replaced “therapeutically feasible concentrations” with “lower concentrations”.

8) Discussion section: "The effects of PUFA analogues on Kv7.1/KCNE1, Cav1.2/β3/α2δ, and Nav1.5/β1 individually are anticipated to have antiarrhythmic effects and would potentially be beneficial for patients with Long QT Syndrome". Basis for this statement?

We have changed the sentence as follows: “The PUFA analogue-induced increases in current seen for Kv7.1/KCNE1 would tend to shorten the AP duration. The PUFA analogue-induced inhibition of Cav1.2/β3/α2δ, and late Nav1.5/β1 currents would also tend to shorten the AP duration. Therefore, all of these effects could potentially be beneficial for patients with Long QT Syndrome. ….Whether these effects are all anti-arrhythmic would have to be tested further in future in vitro and in vivo studies.”

We have shown that PUFA analogues have anti-arrhythmic effects in rat cardiomyocytes (Liin et al., 2015), which we attributed to effects on IKs channels. Alexander Leaf’s group has shown anti-arrhythmic effects of PUFAs like DHA and EPA in animal models due to inhibition of Ca_V_ and Na_V_ (Kang and Leaf, 2000), and we therefore assume that PUFA analogues that inhibit Ca_V_ and Na_V_ may also act anti-arrhythmically and shorten AP duration. We now cite these articles in this paragraph.

9) Note: labeling figures each with its figure # could make the job of the reviewers easier.

Figures are now labeled with numbers.

10) Figure 1: "PUFA analogue, Linoleoyl-taurine, activates Kv7.1/KCNE1 channels through an electrostatic mechanism on voltage sensor and pore."The figure itself does not show that an electrostatic mechanism on voltage sensor and pore is involved.

An extra panel (Figure 1D) is now added that shows an electrostatic mechanism on voltage sensor and pore by PUFA analogues.

11) Figure 2—figure supplement 1: panels C and D are labeled Lin-glycine, while legend states Lin-taurine

Changed to Lin-glycine in legend for panel C and D.

12) Table 1: the Km and A values from many of the fits in Figure 4, Figure 5 and Figure 6 don't seem to be meaningful, as they are underconstrained when dose-responses don't saturate. A statistic for Table I that reflects the error of the fits themselves (e.g. standard deviation representing the confidence of the least mean squared fitting) could be more meaningful.

The inclusion of Km values for the dose responses in the legends for Figure 4 and Figure 5 were done on the request in the previous round of reviews. We agree that these are not well constrained and therefore we did not use the fitted parameters for anything. We instead used the statistics at a single concentration (7 μM) to compare the effects of the different compounds. We have now removed the fitted curves from Figure 4 and Figure 5 and just connected the points with straight lines. Goodness of fits (Adj. R^2^) are now included in Table 1 which refer to data in Figure 6. However, the goodness of fit only tells how well the fitted curve overlaps with the data and does not really tell how unique that fit is. Therefore, we have in Table 1 given estimated Km >values for dose responses that do not clearly saturate. In table 1, N.A. is given for data in which no clear dose-dependent effect is seen.

13) Figure 4 and Figure 5: It is still a bit murky as to how, precisely Gmax/Gmax0 dose-response curves were fit. At low concentrations, the curves don't always plateau at Gmax/Gmax0=1.

In principle, the dose responses should start at 1 (since the data are normalized to the conductance in control solutions), but we have sometimes seen small increases/decreases at low doses (maybe due to run up or run down), so previously we allowed these fits to start at slightly different values of 1 to better fit the low doses. We have now refit the dose responses with a constraint to 1.